# SOLO: A Single Transformer for Scalable Vision-Language Modeling

**Yangyi Chen**,* **Xingyao Wang**\*, **Hao Peng, Heng Ji**
**University of Illinois Urbana-Champaign**
`{yangyic3,xingyao6,haopeng,hengji}@illinois.edu`

**Reviewed on OpenReview:** `https://openreview.net/forum?id=nuzFG0Rbhy&referrer=5B`

## Abstract

We present SOLO, a single transformer for **S**calable visi**O**n-**L**anguage m**O**deling. Current large vision-language models (LVLMs) such as LLaVA mostly employ heterogeneous architectures that connect pre-trained visual encoders with large language models (LLMs) to facilitate visual recognition and complex reasoning. Although achieving remarkable performance with relatively lightweight training, we identify four primary scalability limitations: (1) The visual capacity is constrained by pre-trained visual encoders, which are typically an order of magnitude smaller than LLMs. (2) The heterogeneous architecture complicates the use of established hardware and software infrastructure. (3) Study of scaling laws on such architecture must consider three separate components — visual encoder, connector, and LLMs, which complicates the analysis. (4) The use of existing visual encoders typically requires following a pre-defined specification of image inputs pre-processing, for example, by reshaping inputs to fixed-resolution square images. This inflexibility can create bottlenecks and impede scalability. A unified single Transformer architecture, like SOLO, effectively addresses these scalability concerns in LVLMs; however, its limited adoption in the modern context likely stems from the absence of reliable training recipes that balance both modalities and ensure stable training for billion-scale models. In this paper, we introduce the first open-source training recipe for developing SOLO, an open-source 7B LVLM with the single Transformer architecture using moderate academic resources (8 x A100 80GB GPUs). The training recipe involves initializing from LLMs, sequential pre-training on ImageNet and web-scale data, and instruction fine-tuning on our curated high-quality datasets. On extensive evaluation, SOLO demonstrates performance comparable to LLaVA-v1.5-7B, particularly excelling in visual mathematical reasoning[1].

## 1 Introduction

Large vision-language models (LVLMs) demonstrate remarkable performance on downstream tasks (Li et al., 2023c; Zhu et al., 2023; Liu et al., 2023c; Chen et al., 2023c; Kim & Ji, 2024). They can effectively extract visual information (Wang et al., 2023b) and follow human instructions to generate insightful responses (Li et al., 2023b; Chen et al., 2024d). Two established approaches for vision-language modeling include: (1) Connecting pre-trained visual encoders (Dosovitskiy et al., 2021b; Radford et al., 2021) and large language models (LLMs) (Touvron et al., 2023; Jiang et al., 2023) via a learned projection module that maps the *visual embeddings* to the embedding space of LLMs (Dai et al., 2023; Gao et al., 2023; Liu et al., 2023c), or an intermediate symbolic layer Wang et al. (2024c). (2) Leveraging a pre-trained visual encoder to extract features and aligning feature embeddings with a pre-defined codebook (Esser et al., 2021) to convert each image into a sequence of *discrete visual tokens*, thus enabling LVLMs to process both images and language tokens (Wang et al., 2022b; Peng et al., 2022; Anil et al., 2023; Team, 2024; Diao et al., 2023).

---

*Equal contribution.
[1]The code is made public at `https://github.com/Yangyi-Chen/SOLO`.

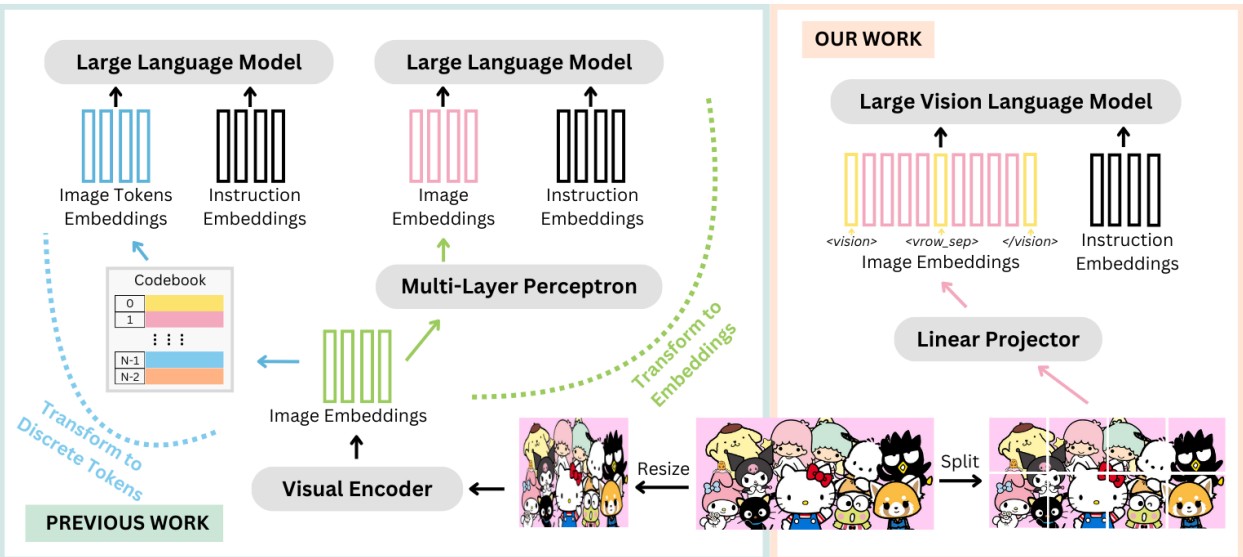

Figure 1: (***Previous work***) The mainstream approaches for vision-language modeling rely on pre-trained visual encoders for visual feature extraction, which exhibits scalability limitations. (***Our work***) We advocate for a unified transformer architecture that processes both images and text, employing a simple linear projection to directly handle raw image pixels. `<vision>`, `</vision>`, and `<vrow_sep>` are special tokens designed explicitly for visual modality encoding.

However, despite their effectiveness, these approaches have limitations that make them hard to scale. We define an architecture as scalable when it exhibits consistent performance improvements as computational resources and training data increase, maintaining a positive scaling law relationship up to practical limits. This is in contrast to architectures that show diminishing returns or performance plateaus at larger scales due to fundamental architectural limitations such as information bottlenecks (*i.e.,* structural limitations in information transmission) in visual perception. The scalability limitations in prevalent LVLMs is evident across four dimensions (§2.1), primarily due to their reliance on a pre-trained visual encoder:

(1) **Constrained visual capabilities:** The visual capacities of a pre-trained vision encoder are largely pre-determined and limited by the data distribution and volume used during pre-training. Due to the significantly smaller size of visual encoders—approximately over ten times smaller than LLMs—they can be a performance bottleneck in solving complex vision-language tasks.

(2) **Challenges in efficient training and deployment:** The heterogeneous architecture of LVLMs with vision encoders complicates adaptation to standard frameworks and hardware optimized for unified Transformer architectures, resulting in reduced computational efficiency in terms of training and inference speed.

(3) **Multiple components complicate the scaling analysis:** The analysis of scaling laws, which are crucial for the development of foundation models, is complicated by the necessity to consider the size of several distinct components independently: the visual encoder, the connector, and the LLMs.

(4) **Limited image pre-processing flexibility:** Most vision encoders pre-define a specification on *how* image inputs should be pre-processed. For example, the widely used visual backbones, such as CLIP-ViT-336 (Radford et al., 2021), require a square image input with a resolution of $336 \times 336$. The inflexibility of image pre-processing can cause bottlenecks that hinder scalability.

To address these limitations, we present SOLO, which employs a **single Transformer architecture for unified and end-to-end vision-language modeling**. SOLO accepts both raw image patches (in *pixels*) and texts as inputs, without using a separate pre-trained vision encoder (Fig. 1). This simplifies the model design and enhances the scalability and adaptability of the LVLM architecture. By simplifying from multi-component LVLM to a single Transformer model, this architecture is unconstrained on the capabilities of visual encoders, easier to train and deploy using existing hardware and software, allows more straightforward scaling law analysis, and can easily scale to image data with diverse resolutions and aspect ratio. SOLO, with a

7-billion parameter count, is initialized from Mistral LLM v0.1 (Jiang et al., 2023) and leverages its extensive pre-trained knowledge.

This modeling strategy is inspired by the foundational modeling framework of VisualBERT (Li et al., 2019) and industry efforts to scale unified LVLMs to the billion-scale (Bavishi et al., 2023). Despite the simplicity and scalability, its limited contemporary adoption can be attributed to the lack of reliable training recipes, as balancing vision and language modalities in unified LVLMs often leads to training divergence. This paper details the first open-source recipe for developing scalable unified LVLMs, using modest academic computational resources, specifically 8 NVIDIA A100 80GB GPUs (§3). Our training recipe involves initializing with pre-trained LLMs, sequential pre-training on ImageNet and web-scale datasets, and instruction fine-tuning on our curated high-quality data mixture.

While still lags behind recent state-of-the-art LVLMs on evaluation benchmarks, SOLO exhibits performance on par with LLaVA-v1.5-7B (§4) and the variant LLaVA-7B* (§6), which is created following our training recipe in the controlled setting. In particular, SOLO distinguishes itself in the domain of visual mathematical reasoning. Further scalability analysis reveals SOLO's better scaling behaviors, inference speed advantages, easier scaling laws analysis, and the scalability and benefits of our flexible image preprocessing pipeline (§6.2). In addition, through comprehensive ablation studies, we validate the design choices of our training recipe. Our empirical results confirm that the sequential pre-training on ImageNet and web-scale datasets and instruction fine-tuning on our carefully curated data mixture are both essential for the training of such single Transformer LVLMs (§5). Interestingly, we find that after removing the first stage of pre-training on ImageNet, the LVLM will produce outputs of drastically different quality while exhibiting similar image-conditioned language modeling loss (§5.1, Fig. 3).

## 2 Tackling Scalability Limitations via Integrated Architectures

### 2.1 Scalability Limitations in Existing LVLMs

The scalability constraints of existing LVLMs are currently articulated from four critical perspectives that limit their efficiency in utilizing expanded computational resources and larger datasets due to the bottlenecks in the system design:

**Fixed and Constrained Visual Capabilities**   The fixed nature of visual encoders severely limits the adaptability of LVLMs to novel visual data distribution and more complex vision-language tasks since these encoders are trained on specific distributions and training objectives. Current approaches address this issue by continuing the training of visual encoders (Bai et al., 2023) or by integrating features derived from various visual encoders (Lin et al., 2023). Nonetheless, the scope of data used for continued pre-training is substantially less than that used initially, which only marginally enhances encoder adaptability, and employing multiple encoders complicates the process of image feature extraction, thereby impeding the scalability of LVLMs. Moreover, the smaller scale of visual encoders compared to LLMs frequently results in the visual understanding component becoming a bottleneck. Consequently, visual representation learning is limited to the smaller visual encoders, hindering the full utilization of LLM capabilities in existing LVLMs.

**Challenges in Efficient Training and Deployment**   The heterogeneous architecture with multiple components complicates the implementation of machine learning systems for efficient training and deployments in terms of speed. (1) **Training challenge**: At large training scale (*i.e.,* multi-node clusters), it is necessary to distribute not only the Transformer-based LLMs but also the vision model and the MLP connector across multiple devices, employing techniques such as tensor and pipeline parallelism. Thus, prevalent LVLMs cannot directly use existing industry-grade training frameworks optimized for the Transformer architecture (Shoeybi et al., 2019; Cano et al., 2023), thus necessitating the development of new tensor-sharding mechanisms. In addition, AI alignment typically employs algorithms such as Proximal Policy Optimization (PPO) (Schulman et al., 2017), which necessitate simultaneously maintaining multiple models (*e.g.,* reward and critic models) in GPU memory and cause difficulty in the algorithm implementations for heterogeneous architectures. (2) **Deployment**: The heterogeneous architecture complicates the deployment process due to similar model and tensor sharding challenges described above. Consequently, this hampers the large-scale services of existing

LVLMs. Moreover, existing specialized AI chips (Techcrunch) and inference libraries, such as vLLM (Kwon et al., 2023) and MLC-LLM (team, 2023), are mostly optimized for Transformer architectures, presenting significant challenges in the deployment of these models on end devices.

**Multiple Components Complicate the Scaling Analysis** The complexity introduced by the multiple components of LVLMs is a significant barrier to understanding and improving these systems. Each component —the visual encoder, the connector, and the language models—operates with its own parameters and training strategies (Radford et al., 2021; 2019; Brown et al., 2020a), which can lead to a lack of cohesion in the overall model behavior. Scaling laws are crucial for guiding the development of large foundational models by forecasting the performance of a target model using data from several sampled models that are significantly smaller in sizes (Kaplan et al., 2020; Bahri et al., 2021). However, applying these approaches to existing LVLMs requires simultaneous consideration and scaling of various components, hereby increasing complexity.

**Limited Image Pre-Processing Flexibility** The strict requirements for image pre-processing imposed by the specifications of visual encoders may create bottlenecks that hinder the scalability. For instance, the requirement of a consistent input resolution can make it difficult to process images that are naturally high-resolution or have non-standard aspect ratios without compromising on the quality or representational fidelity of the input. Current mitigation strategies involve splitting the original image into multiple sub-images, independently extracting visual features from each sub-image using pre-trained visual encoders and subsequently aggregating the representation embeddings (Xu et al., 2024a; Liu et al., 2024a; Dong et al., 2024). However, these approaches seem ad-hoc and can be suboptimal as the visual backbone is not pre-trained to handle these inputs, potentially impacting the effective handling of these high-resolution images.

## 2.2 Unified Vision-Language Modeling with Integrated Architectures

We revisit the foundational modeling framework of VisualBERT (Li et al., 2019), initially proposed in the early stages of research on pre-trained vision-language models. The key idea is to use one single Transformer, initialized from BERT (Devlin et al., 2018) in VisualBERT, to uniformly process the image patches and language tokens. Fuyu-8B exemplifies the industry's effort to scale this modeling approach (Li et al., 2019) to billion-scale models (Bavishi et al., 2023). However, the limited widespread implementation of this unified architecture may be due to the lack of an established training recipe, as only the pre-trained model is released by Bavishi et al. (2023) without training details. Training such unified LVLMs presents significant challenges in balancing the two modalities and maintaining stable training, for which clear solutions are currently lacking. In this paper, we present `SOLO` with full details of its unified and integrated architecture design and training recipe.

## 3 `SOLO`: Scalable Vision-Language Modeling

`SOLO` consolidates image and language capabilities into a single model, enables data-driven determination of visual representations and parameter allocation across visual and language modalities, simplifies the scaling laws analysis, and allows it to handle high-resolution images and those with uncommon aspect ratios flexibly. For large-scale training (§3.2), `SOLO` also seamlessly integrates with established software frameworks for large-scale Transformer pre-training (Shoeybi et al., 2019).

## 3.1 Model Architecture

The architecture of `SOLO` is shown in Fig. 1, which diverges from earlier models primarily in the extraction of visual features. Instead of resizing the image into a fixed resolution adapted to the pre-trained image encoders, `SOLO` keeps their original resolutions and aspect ratios. The feature extraction involves splitting the image into patches with a pre-defined size. Through a trainable linear projection, these raw image patches (in *pixels*) are transformed to obtain continuous embeddings that represent the visual features of the images. Thus, we can integrate image and language processing within a single model. We maintain a list of special tokens designed explicitly for visual modality encoding: `<vision>` and `</vision>` tokens mark the beginning and end of a span of image patches respectively; `<vrow_sep>` acts as a row separator within the image patches and helps the model distinguish between different rows of image patches, aiding in structured visual understanding.

Table 1: Summary of datasets used in the three stages of pre-training. Each image patch counts as a vision token. Number of *instances* is calculated after packing them into sequences with 32K length.

| Training Stage | Dataset | #Instances | #Image | #Token | #Text Tokens | #Vision Tokens |
|---|---|---|---|---|---|---|
| **Stage-1** | ImageNet21K (Ridnik et al., 2021b) | 74,283 | 13,151,276 | 2,423,203,108 | 212,745,573 | 2,210,457,535 |
| | SlimPajama, *subset* (Soboleva et al., 2023) | 120,839 | 0 | 4,340,877,587 | 4,340,877,587 | 0 |
| | **Total** | 195,122 | - | 6,764,080,695 | (67.32%) 4,553,623,160 | (32.68%) 2,210,457,535 |
| **Stage-2** | Capfusion (subset) (Yu et al., 2024) | 204,978 | 23,681,864 | 6,664,351,863 | 1,172,726,505 | 1,172,726,505 |
| | Websight (Laurençon et al., 2024) | 71,579 | 1,922,671 | 2,300,945,215 | 1,087,060,511 | 1,213,884,704 |
| | CC3M (Sharma et al., 2018b) | 32,760 | 2,331,439 | 1,064,477,314 | 76,092,147 | 988,385,167 |
| | Detailed Captions (lz) | 6,225 | 368,767 | 202,016,770 | 44,788,200 | 157,228,570 |
| | LLaVAR (Zhang et al., 2023b) | 3,602 | 422,315 | 117,448,784 | 31,390,556 | 86,058,228 |
| | DVQA (Kafle et al., 2018) | 2,917 | 200,000 | 94,853,796 | 55,653,796 | 39,200,000 |
| | OCR-VQA (Mishra et al., 2019) | 1,593 | 165,746 | 51,920,705 | 21,161,018 | 30,759,687 |
| | FigureQA (Kahou et al., 2017) | 1,526 | 100,000 | 49,586,305 | 24,803,256 | 24,783,049 |
| | SlimPajama, *a different subset* (Soboleva et al., 2023) | 120,385 | 0 | 4,300,998,161 | 4,300,998,161 | 0 |
| | **Total** | 445,565 | - | 14,846,598,913 | (45.90%) 6,814,674,150 | (54.10%) 8,031,924,763 |
| **Stage-3** | ALLaVA-LAION (Chen et al., 2024a) | 13,725 | 438,992 | 442,509,490 | 176,660,898 | 265,848,592 |
| | ALLaVA-VLFLAN (Xu et al., 2024b) | 4,469 | 207,549 | 144,577,377 | 77,835,919 | 66,741,458 |
| | LLaVAR (Zhang et al., 2023b) | 3,602 | 422,315 | 117,448,784 | 31,390,556 | 86,058,228 |
| | DVQA (Kafle et al., 2018) | 2,917 | 200,000 | 94,853,796 | 55,653,796 | 39,200,000 |
| | FigureQA (Kahou et al., 2017) | 1,526 | 100,000 | 49,586,305 | 24,803,256 | 24,783,049 |
| | SlimPajama, *a different subset* (Soboleva et al., 2023) | 12,085 | 0 | 430,688,442 | 430,688,442 | 0 |
| | **Total** | 38,324 | - | 1,279,664,194 | (62.28%) 797,032,867 | (37.72%) 482,631,327 |

Formally, we define the patch size $P$ and the maximal resolution $M$. For an image of dimension size $(W, H)$, it is resized to $(W', H')$ to ensure divisibility by $P$. Fig. 2 details the resizing process, which adjusts the $W$ and $H$ to the nearest multiples of $P$ while preserving the original aspect ratio to the extent possible and complying with the constraints imposed by $M$. Subsequently, the image is divided into $N$ patches, where $N = (W'/P) \times (H'/P)$, each with dimensions $P \times P \times 3$. A trainable linear projector then maps each patch from a flattened $P \times P \times 3$ vector to an output dimension compatible with the embedding space of LLMs, extracting $N$ embeddings as the image's feature representation. These visual embeddings, along with special visual modality tokens and embeddings of the text tokens, are concatenated and processed through a single Transformer, facilitating unified vision-language modeling. Notably, compared to prevalent LVLMs, this modeling strategy facilitates

```
PATCH_SIZE = 32
MAX_RESOLUTION = 1024  # 32 x 32

def get_resize_output_image_size(image_size):
    l1, l2 = image_size
    if l2 <= l1:
        short, long = l2, l1
    else:
        short, long = l1, l2

    requested_new_long = min(
        int(long / PATCH_SIZE + 1) * PATCH_SIZE,
        MAX_RESOLUTION
    )
    new_long = requested_new_long
    new_short = int(new_long * short / long)
    new_short = int(new_short / PATCH_SIZE + 1)
        * PATCH_SIZE

    if l2 <= l1:
        return new_long, new_short
    else:
        return new_short, new_long
```

Figure 2: The input image resize algorithm to maintain the aspect ratio.

a much earlier fusion of visual and language modalities, allowing LVLMs to extract relevant information conditioned on the given instructions. In our implementation, we initialize `SOLO` from the Mistral-7B-v0.1 base LLM. The max resolution $M$ of processed images is set as 1024. The patch size $P$ is set as 32.

## 3.2 Training Recipe

We describe our approach for training unified billion-scale LVLMs, including pre-training (§3.2.1) and instruction fine-tuning (§3.2.2). For both stages, we optimize exclusively the language modeling loss on natural language tokens, without optimizing loss on image patches and special image tokens (*e.g.,* <vision>). We substantiate the essential ingredients in our recipe in §5.

### 3.2.1 Pre-Training

We introduce a three-stage pre-training curriculum that progressively enhances the visual capabilities of LVLMs while preserving their fundamental language capabilities. We present datasets and their statistics for each stage in Tab. 1.

**Stage-1 ImageNet Pre-Training for Initialization** We leverage ImageNet21K (Ridnik et al., 2021a), encompassing a broad spectrum of fine-grained visual categories, for the initial pre-training stage. In this process, we train `SOLO` to predict *only* fine-grained labels in natural language tokens (class name of images,

*e.g.,* "golden retriever") conditioned on the image patches, thereby developing visual representations that initialize subsequent pre-training runs. In §5, we demonstrate the critical role of this stage in training unified LVLMs: when this stage is removed, the LVLM pre-trained on web-scale data from stage 2 (*e.g.,* captioning) failed to generate meaningful captions (Fig. 4).

**Stage-2 Pre-Training on Web-Scale Data**  ImageNet21K, composed chiefly of visual concept data annotated by humans, faces scalability constraints in both knowledge breadth and data volume. In Stage-2, we scale up the pre-training data to encompass web-scale data, primarily consisting of image-caption pairs from sources like Capfusion (Yu et al., 2024) and CC3M (Sharma et al., 2018a). Additionally, we include synthetically generated web pages with associated HTML code from Websight (Laurençon et al., 2024) to improve OCR performance, and we also include a small set of supervised datasets to improve the data diversity. In this stage, the language modeling loss is applied uniformly across all language tokens, encompassing captions, HTML code, and questions and responses within the supervised datasets.

**Stage-3 Annealing**  Following MiniCPM (Hu et al., 2024a), we perform a final annealing stage to conclude the pre-training. In this stage, we incorporate a limited selection of supervised datasets–either down-sampled or omitted from the instruction fine-tuning dataset mixture (*e.g.,* ALLaVA, Chen et al. 2024a)–to prime SOLO for the subsequent instruction fine-tuning stage. The primary purpose of this stage is to transition SOLO from a noisy web data to being trained on high-quality data mixtures.

**Balancing Text and Vision Capability Through Language Corpus Blending**  Initiating with a base LLM and performing full-parameters training necessitates carefully preserving its inherent language comprehension abilities while performing image representation learning since most real-world vision-language tasks require text-only capabilities such as instruction comprehension and complex reasoning. At each stage of SOLO's pre-training, we mix in a non-trivial proportion of text-only pre-training data (SlimPajama, Soboleva et al. 2023) to maintain the text capability. We present more empirical results on how data mixture affects image and text loss trade-offs in §7.2.

**Pre-Training Infrastructure**  We modify the standard Megatron-LLM (Cano et al., 2023) to support arbitrary image patch inputs. We use one node with 8 NVIDIA A100 80G GPU for pre-training. We use 2-way tensor parallelism (Shoeybi et al., 2019) and 4-way data parallelism for training. Following Shoeybi et al. (2019), we adopt distributed optimizer to shard optimizer states across different GPUs for memory efficiency. In our test on an 8xA100 server with identical image-caption pre-training, Megatron achieves 20K tokens per second, 67% higher than DeepSpeed, making it more suitable for our pre-training requirements.

**Training Hyperparameter**  We use a global batch size of 128 examples (*i.e.,* 4M tokens) and each pre-training example is packed to $32,768$ tokens. We adopt a learning rate of 5e-5 with cosine decay to a minimum learning rate of 5e-6 and warm up for 200 steps. We use weight decay of 0.1. For training efficiency, we pack shorter sequences into one longer sequence and re-adjust the attention mask to make sure tokens from different examples cannot attend to each other. The training process consists of 1525 steps in Stage 1, 3480 steps in Stage 2, and 300 steps in Stage 3.

### 3.2.2 Instruction Fine-Tuning

**Dataset Curation**  We meticulously select a diverse range of supervised datasets to perform instruction fine-tuning, aiming to enhance their performance across various domains of vision-language tasks. Our dataset selection strategy is based on the empirical analysis derived in Laurençon et al. (2024); Lin et al. (2024); Lu et al. (2024), and is mainly driven by the objective to cover a comprehensive range of data types, including language-only data, detailed image captions, scientific documents, tables, documents, charts, OCR and text-rich images, and general visual question-answering (VQA) tasks. In §A, we present datasets and their statistics in Tab. 7 and more details regarding data curation.

**Implementation Details**  We utilize DeepSpeed (Rasley et al., 2020), as implemented in Accelerate (Gugger et al., 2022), for instruction fine-tuning. The choice to use Accelerate over Megatron for fine-tuning is

based on practical efficiency. Accelerate enables quick experimentation with data mixtures by modifying a few lines of code, while Megatron demands extensive preprocessing for each run, complicating ablation studies. Additionally, Megatron's complex, multi-layered implementation hinders customization, presenting challenges in potential further extension, such as introducing advanced methods like RLHF for alignment. For hyperparameters, the global batch size is configured at 640, with a weight decay parameter of 0.1. We train for 1 epoch with a maximum learning rate of 1e-5, which follows a linear warm-up phase and transitions to a cosine decay schedule.

## 4 Comparison to Existing LVLMs

### 4.1 Model

We select various open-source LVLMs for comparison to better understand the capabilities of `SOLO`. Based on the release time and capabilities of LVLMs, we select 3 groups of LVLMs to better understand the current development phase of `SOLO`. Level-1 LVLMs represent the pioneering generation, which initiate the integration of visual encoders with pre-trained LLMs, with releases prior to October 2023. Level-2 LVLMs, released before early 2024, typically feature a more refined selection of instruction fine-tuning data to enhance performance. Level-3 marks the state-of-the-art (SoTA) LVLMs, released within the last five months, incorporating advanced training recipes, superior LLM backbones, and support for high-resolution images.

- **Level-1**: (1) OpenFlamingo v2 (Awadalla et al., 2023), (2) MiniGPT4 v2 (Chen et al., 2023a), (3) VisualGLM (Du et al., 2022), (4) InstructBLIP (Dai et al., 2023), (5) LLaVA v1 (Liu et al., 2023a).
- **Level-2**: (6) LLaVA v1.5 (Liu et al., 2024a), (7) mPLUG-Owl v2 (Ye et al., 2024), (8) InternLM-XComposer (Zhang et al., 2023a), (9) MiniCPM-v1 (Hu et al., 2023),
- **Level-3**: (10) Monkey (Li et al., 2024b). (11) LLaVA-NEXT (Liu et al., 2024b), (12) MiniCPM-v2 (Hu et al., 2024b), (13) DeepSeek-VL (Lu et al., 2024).

Each LVLM may have multiple variants based on different LLM sizes and architectures. If possible, we opt for the variant equipped with a 7B Mistral LLM. For the remaining LVLMs, we select the variant whose configuration most closely aligns with our specifications (Mistral-7B-LLM). We directly present the evaluation results of existing LVLMs from the leaderboard (OpenCompass) when available, to ensure a fair comparison.

### 4.2 Benchmarks

We select a wide range of benchmarks, encompassing both general vision-language tasks and specific task-oriented datasets, for evaluation and analysis. For general vision-language capability evaluation, we choose MMStar (Chen et al., 2024b), MME (Fu et al., 2024), and SEED-Bench (Li et al., 2024a). Specifically, MMStar measures elite vision-indispensable capabilities, MME measures both the perception and cognition capabilities, and SEED-Bench covers 12 evaluation dimensions covering various aspects of LVLMs capabilities. For scientific document understanding, we choose AI2D (Kembhavi et al., 2016) and ScienceQA (Lu et al., 2022a). For visual mathematical reasoning, we choose MathVista (Lu et al., 2023). We adopt VLMEvalKit (Contributors, 2023; Duan et al., 2024) to perform the unified evaluation.

### 4.3 Results

The experimental results are shown in Tab. 2. We find that `SOLO` significantly outperforms Level-1 LVLMs and also performs comparably to Level-2 LVLMs, despite slightly underperforming Level-3 LVLMs. Furthermore, `SOLO` excels in task-oriented benchmarks, especially in areas requiring scientific knowledge and mathematical reasoning, due to its successful integration of image representation and complex reasoning within a single unified model. Overall, while `SOLO` does not yet meet the SoTA performance of the leading LVLMs (Level-3) within the prevalent multi-component LVLM framework, it marks a substantial progression in unified vision-language modeling. It is important to consider that `SOLO` is trained using limited academic resources, specifically 8 A100 GPUs. This is in sharp contrast with the resources used to produce SoTA LVLMs. For example, the technical report of DeepSeek-VL (Lu et al., 2024) mentions that DeepSeek-VL-7B is trained on a cluster of 64 nodes, each comprising 8 Nvidia A100 GPUs, totaling 512 GPUs—this represents a resource

Table 2: The main experimental results of `SOLO`. We compare with LVLMs with diverse capabilities, released at different times and categorized into three levels. `SOLO` aligns with the second level of prior LVLMs advancements focusing on LLaVA-Style modeling, and distinguishes itself in visual mathematical reasoning. (C) denotes the visual encoders from CLIP.

| Level | Model | Visual | Language | MMStar | MME | SEED | ScienceQA | MathVista | AI2D |
|---|---|---|---|---|---|---|---|---|---|
| Level-1 | OpenFlamingo v2 | (C) ViT-L/14 | MPT-7B | 26.9 | 607.2 | 28.8 | 44.8 | 18.6 | 31.7 |
| | MiniGPT-4-v2 | EVA-G | Llama2-13B | 21.3 | 968.4 | 29.4 | 54.7 | 23.1 | 30.5 |
| | VisualGLM | EVA-CLIP | ChatGLM-6B | 5.9 | 738.1 | 47.0 | 56.1 | 21.9 | 41.2 |
| | InstructBLIP | EVA-G | Vicuna-7B | 32.7 | 1391.4 | 44.5 | 54.1 | 24.4 | 40.6 |
| | LLaVA-v1-7b | (C) ViT-L/14 | Llama-7B | 27.1 | 1075.5 | 50.4 | 61.8 | 25.2 | 48.3 |
| Level-2 | LLaVA-v1.5 7b | (C) ViT-L/14 | Vicuna-V1.5-7B | 33.1 | 1808.4 | 65.8 | 69.2 | 25.6 | 55.5 |
| | mPLUG-OWL v2 | (C) ViT-L/14 | Llama2-7B | 34.8 | 1786.4 | 64.5 | 69.5 | 25.4 | 55.7 |
| | XComposer | EVA-G | InternLM-7B | 6.9 | 1874.2 | 66.1 | 89.8 | 29.8 | 56.9 |
| | MiniCPM-V | SigLIP-400M | MiniCPM-2.4B | 38.6 | 1650.2 | 65.6 | 77.0 | 30.6 | 56.3 |
| Level-3 | Monkey | ViT-BigHuge | Qwen-7B | 37.0 | 1759.9 | 64.3 | 72.1 | 33.5 | 62.5 |
| | LLaVA-Next | (C) ViT-L/14 | Mistral-7B | 38.4 | 1821.2 | 72.4 | 73.0 | 34.6 | 69.0 |
| | MiniCPM-v2 | SigLIP-400M | MiniCPM-2.4B | 39.1 | 1808.2 | 67.1 | 80.7 | 39.8 | 62.9 |
| | DeepSeek-VL | Hybrid | DeepSeek-7B | 40.5 | 1765.4 | 70.1 | 80.9 | 36.9 | 65.3 |
| Ours | `SOLO` | Mistral-7B | | 35.5 | 1260.0 | 64.4 | 73.3 | 34.4 | 61.4 |

scale 64 times greater than that used for `SOLO`. `SOLO` serves as a pivotal model, showcasing the scientific value of training LVLMs with unified architectures. This establishes `SOLO` as a viable candidate for future developments aimed at closing the performance gap with SoTA LVLMs, with more flexibility and scalability by avoiding issues in prior architectures (§2.1).

## 5 Validating Key Ingredients in Our Recipe

### 5.1 LVLMs Generate Meaningless Captions without Stage-1 Pre-training

We assess the necessity of Stage-1 pre-training by comparing the Stage-2 LVLM checkpoints *with* and *without* undergoing Stage-1 ImageNet pre-training. In Fig. 3, we observe that these two variants overall achieve similar pre-training loss curves on vision-language modeling and (text) language modeling.

**Select Checkpoints for Comparison** We select two checkpoints for comparison: one using caption-only pre-training (Stage-2 only) and the other utilizing `SOLO`'s two-stage pre-training, both of which achieve an equivalent vision-language modeling loss of 2.1.

**Qualitative Comparison** We *randomly* select one example for qualitative analysis (in Fig. 4). Despite the equivalent loss of the selected checkpoints in Fig. 3, we find that without ImageNet pre-training (Stage-2 only), the model generates irrelevant and meaningless image captions, indicates a training divergence.

**Quantitative Comparison** We perform a quantitative comparison on the same two checkpoints by training them on the instruction fine-tuning data mixture for 800 steps (see Fig. 5a). Compared to the two-stage pre-trained `SOLO`, we observe a performance degradation across multiple benchmarks on the checkpoint *without* Stage-1 pre-training, further validating the importance of the first stage.

**Discussion** We hypothesize that discrepancies between a model's vision and language capabilities can lead to the observed behaviors. Specifically, when there is a significant imbalance—such as with the Mistral 7B model, which possesses advanced language abilities but lacks vision understanding—the model may reduce loss by replicating caption patterns, including redundant text tokens irrelevant to the visual content. For instance, in a caption like "This is a dog", the essential element is "dog". Focusing solely on minimizing language modeling loss without a robust initialized vision representation may lead the model to favor generic phrases like "This is a" over the more discriminative "dog". This is because the former includes more tokens, disproportionately influencing the overall language modeling loss. Pre-training on ImageNet at Stage 1, which emphasizes predicting only the "dog" token, helps the model develop a solid visual representation, effectively narrowing the gap between vision and language capabilities and mitigating this issue. In addition, the results indicate that pre-training loss on vision-language data does not reliably indicate the performance of

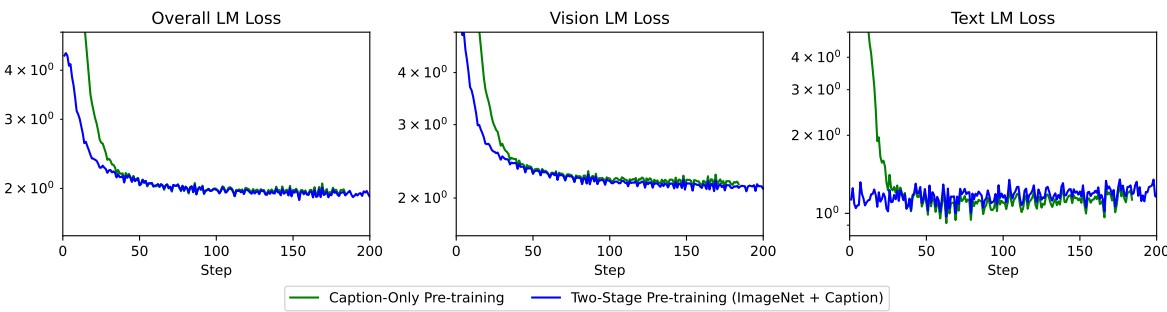

Figure 3: Image captioning loss using two differently initialized checkpoints: (**1**) caption-only pre-training (green) initialized from the LLM; (**2**) two-stage pre-training (blue) initialized from the Stage-1 ImageNet pre-trained LVLM.

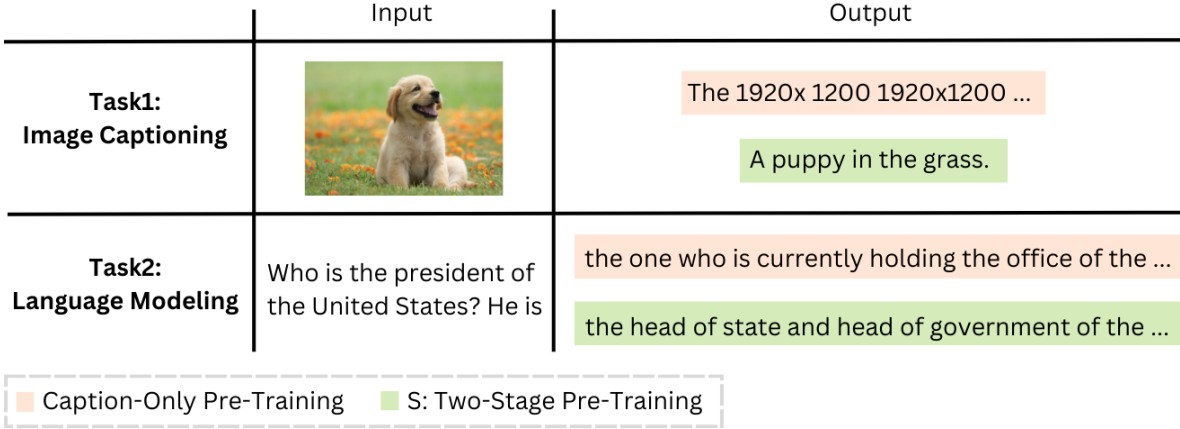

Figure 4: Qualitative analysis of caption-only pre-training and SOLO's two-stage pre-training. Comparisons are made on two checkpoints with comparable vision-language modeling loss (*i.e.,* 2.1). Specifically, we select the caption-only checkpoint at pre-training step 150, and SOLO at step 100.

LVLMs. Detailed analysis are provided in §7.1, which also demonstrates that training loss on the instruction fine-tuning data mixture is similarly unreliable as an indicator.

## 5.2 Stage-2 Pre-Training on Web-Scale Data

**Stage-2 Pre-Training Improves Performance on Top of Stage-1** We verify the effectiveness of Stage-2 pre-training on web-scale data by comparing the performance of two LVLMs. Each model is fine-tuned for 800 steps using the same instruction fine-tuning data mixture but initialized differently—one from the end of Stage 1 and the other from Stage 2. In Fig. 5b, we observe significant improvement on all evaluation datasets after pre-training on web-scale data, showing the substantial advantages of Stage 2 pre-training compared to solely using ImageNet data (Stage 1 only).

**Combining ImageNet and Captioning at Stage-2 Hurts Performance** In addition, it is pertinent to ask whether ImageNet21K data can be combined with web-scale data for Stage 2. We include an ablation with SOLO trained on ImageNet21K and the web-scale data included in the second stage. Fig. 6 illustrates the training curves for comparison. The results suggest that while ImageNet pre-training effectively establishes an initial visual representation, it may not be optimal for subsequent Stage-2 pre-training on web-scale data, as it potentially impedes the optimization of vision-language modeling on image captions (*i.e.,* vision language modeling loss stop improving). This discrepancy may arise from the divergence in image classification and captioning capabilities; the former is emphasized in the first stage. This two-stage approach aligns with the principles of continual curriculum learning, where the model must maintain proficiency in familiar tasks while integrating new ones. This conclusion is also supported by our evaluation of downstream tasks (see

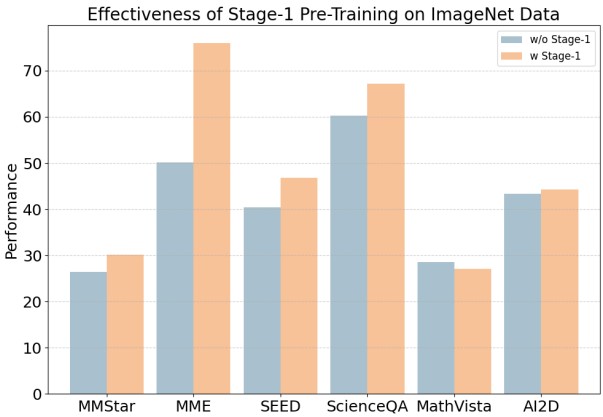

(a) The effectiveness of Stage-1 pre-training on ImageNet data for initialization.

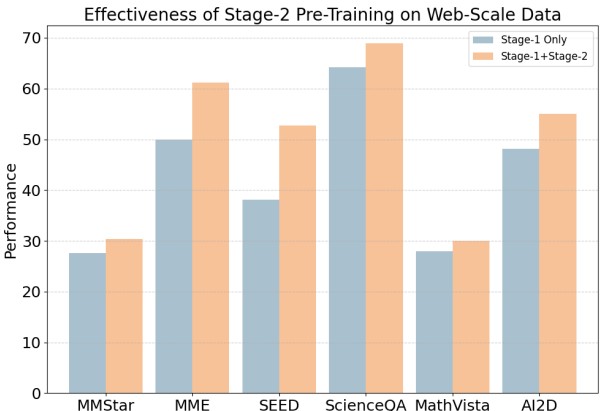

(b) The effectiveness of Stage-2 pre-training on web-scale data for knowledge breadth and data volume.

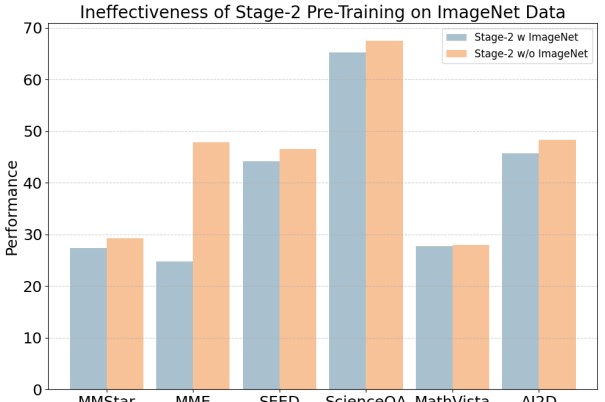

(c) The ineffectiveness of including ImageNet data in Stage-2 pre-training.

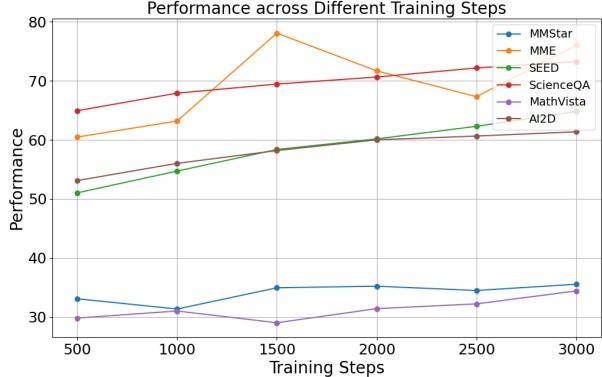

(d) The performance across training steps on the fine-tuning dataset.

Figure 5: The evaluation performance of various ablations to validate key ingredients of our recipe. The MME scores are normalized for better illustration.

Fig. 5c). We train two different checkpoints with and without ImageNet21K data in the second stage on the instruction fine-tuning data mixture (§3.2.2) for 800 steps. Note that we select two checkpoints with comparable vision-language modeling losses for analysis. The results indicate that incorporating ImageNet21K data in the second stage may detrimentally impact overall performance by inhibiting adaptation to and learning from web-scale data.

## 5.3 Performance Boost via Instruction Fine-Tuning

We evaluate the performance of SOLO on different training steps throughout the instruction fine-tuning stage (see Fig. 5d). The results indicate a consistent improvement in SOLO's performance with prolonged training on the fine-tuning dataset, although the MME scores exhibit some fluctuations. This outcome contrasts with the findings of Liu et al. (2024a), where the performance quickly plateaus upon training with a limited subset of the fine-tuning dataset. This illustrates the increased scalability of SOLO during the instruction fine-tuning stage, suggesting that acquiring additional high-quality supervised datasets for fine-tuning could consistently enhance performance.

## 5.4 Additional Validation Experiments

We present the results to demonstrate the effectiveness of Stage-3 annealing in §B and to validate the curated data mixture for instruction fine-tuning in §C.

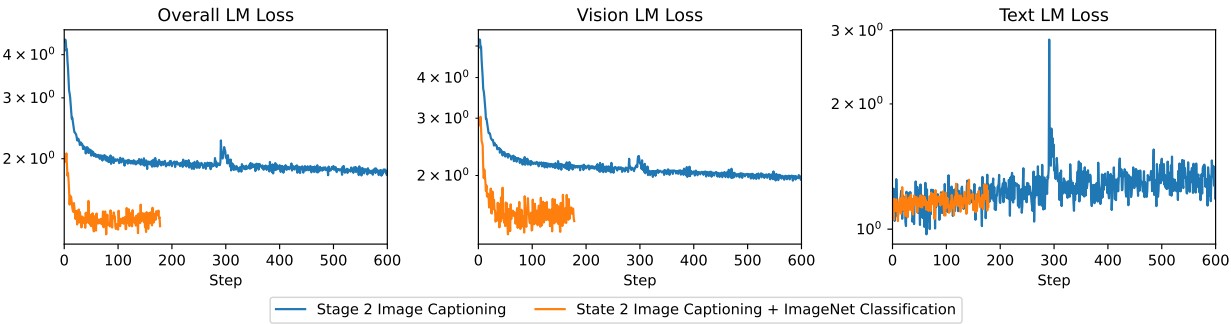

Figure 6: Stage-2 language modeling loss when trained on (**1**) Image captioning objective only (blue); (**2**) Image captioning objective *and* image classification objective used in Stage-1 (orange).

## 6 Further Analysis

### 6.1 Controlled Analysis of Fuyu and LLaVA

We conduct a controlled analysis to compare `SOLO` with LLaVA and Fuyu (see Fig. 7). `SOLO` with our training recipe consistently outperforms Fuyu-8B, which adopts the same unified modeling strategy, across all evaluation benchmarks. To facilitate a controlled comparison with LLaVA, we develop LLaVA-7B*, which integrates CLIP-ViT-336 and Mistral-7B-base-v0.1, utilizing our specific training procedure and data. The results reveal that LLaVA-7B* achieves performance similar with LLaVA-v1.5-7B (Liu et al., 2024a), indicating that our training recipe, which utilizes large-scale datasets and extensive training, may not significantly impact LLaVA-style LVLMs. Notably, while LLaVA-7B* excels in general visual-language tasks, `SOLO` demonstrates superior capabilities in visual mathematical reasoning, with overall performance being similar.

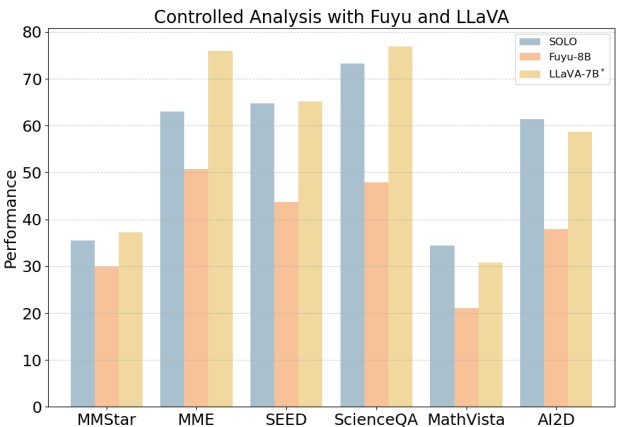

Figure 7: The controlled analysis of Fuyu-8B and LLaVA-7B.

### 6.2 Scalability Analysis

**Superior Scaling Properties of `SOLO`** We demonstrate that existing LVLMs with heterogeneous architectures exhibit diminishing returns despite increases in high-quality instruction fine-tuning data while `SOLO` shows better scaling behaviors. We perform instruction fine-tuning on pre-trained LLaVA obtained in §6.1 and compare its scaling behaviors with `SOLO` by measuring the performance improvement per training token. For evaluation, we fine-tune both models for 50 steps as their initial checkpoints, as their pre-trained versions are not effective at following instructions. Given that both models are fine-tuned on the same dataset, we can directly compare their average improvement in benchmark performance per training token during training. In the implementation, we measure performance improvement every 500 steps, normalized by the number of training tokens encountered during those steps, and average this to calculate the final metric. Fig. 8 shows that `SOLO` outperforms mLLaVA on the "performance improvement per token" metric across all evaluation benchmarks. This suggests `SOLO` benefits more from high-quality instruction fine-tuning data and demonstrates better scalability, indicating that its performance could further be improved more compared to mLLaVA with more data.

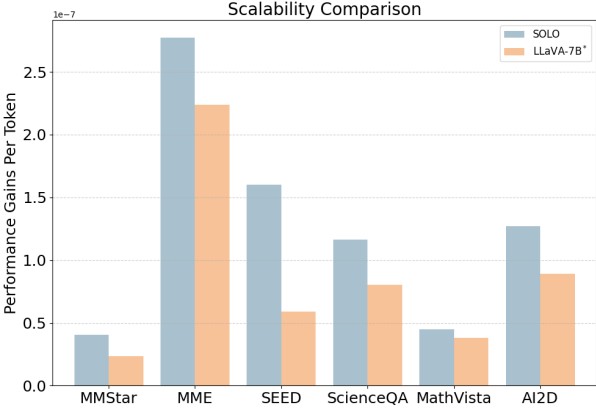

Figure 8: We compare the scaling behaviors of `SOLO` and LLaVA by measuring the improvement on benchmark performance per token.

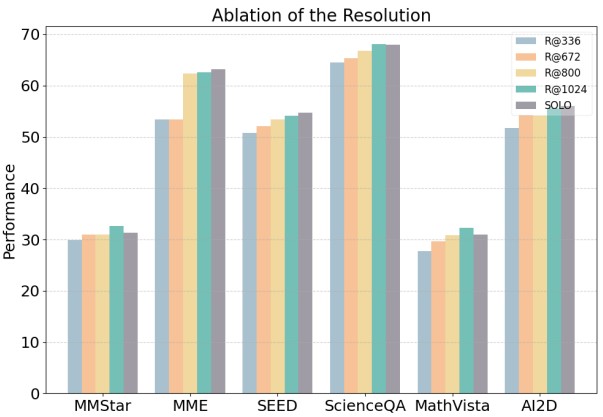

Figure 9: The performance of `SOLO` when trained and tested on different resolutions of images. R@X denotes the resolution of X.

**`SOLO` Exhibits Training and Inference Speed Advantage** We measure the training speed of SOLO compared to LVLMs with heterogeneous architectures by measuring throughput. Using the same 8xA100 server, we compare the number of tokens processed per second during SOLO training and the official LLaVA implementation (Liu et al., 2023a). Our SOLO implementation achieves a throughput of 20K tokens per second, while LLaVA reaches 10.5K tokens per second, highlighting

Table 3: Inference latency comparison.

| Model | Inference Latency (seconds) |
|---|---|
| LLaVA-Next | 0.0641 |
| LLaVA-Interleave | 0.0691 |
| Qwen2-VL | 0.0588 |
| SOLO | **0.0235** |

SOLO's significant advantage in training speed. We also measure the inference speed of `SOLO` compared to several LVLMs with heterogeneous architectures, including LLaVA-Next, LLaVA-Interleave, and Qwen-VL. The evaluation is conducted on a single A100 GPU using 10,000 COCO images and a consistent prompt: "Generate the Caption for the <image>". All inference code is implemented using HuggingFace, and the inference latency is measured from the input being provided to the model and the output of the first token for fair comparison among all models since they may generate free-form responses in different lengths. The results, shown in Tab. 3, indicate that `SOLO` consistently outperforms other LVLMs with heterogeneous architectures by a significant margin, demonstrating its clear advantage in inference speed and large-scale development.

**`SOLO` Facilitates Easier Scaling Laws Analysis** We conduct a simplified scaling laws experiment to show that the performance gains from scaling up data for `SOLO` is more predictable compared to LLaVA-Style LVLMs. Specifically, we follow Kaplan et al. (2020) to fit the analytical function that LVLMs trained with a limited dataset (instruction fine-tuning in our case):

Table 4: Predictability of the performance gains comparison between SOLO and mLLAVA on various benchmarks.

| $R^2$ | MMStar | MME | SEEDBENCH | ScienceQA | MathVista | AI2D |
|---|---|---|---|---|---|---|
| mLLAVA | 40.32 | 92.52 | 92.89 | 89.25 | 62.73 | 83.29 |
| SOLO | 88.75 | 92.36 | 98.59 | 97.75 | 84.70 | 96.55 |

$$L(D) = \left(\frac{D_c}{D}\right)^{\alpha_D},$$

where $D_c$ and $\alpha_D$ are constants to be estimated, $D$ is the number of training tokens, $L(D)$ is the benchmark performance in our case. We use data points from the instruction fine-tuning of `SOLO` and mLLaVA to fit this function and report the coefficient of determination $R^2$, which reflects the quality of the fit and is equivalent to the predictability of performance. The results below demonstrate that `SOLO`'s performance is more predictable compared to mLLaVA, indicating that `SOLO` facilitates easier scaling laws analysis. We further provide more discussion about the better scalability of `SOLO` in §7.3.

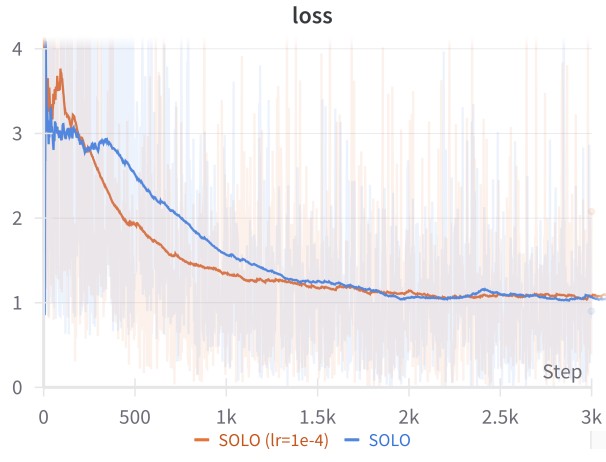
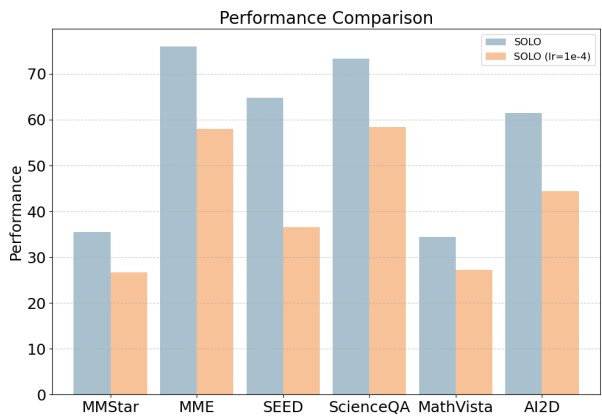

(a) The training curves of two variants.

(b) The downstream performance of two variants.

Figure 10: The training curves and downstream performance evaluation of two variants of `SOLO`. We find that they show significant performance differences although achieving a similar loss on the instruction fine-tuning data mixture.

**`SOLO` Demonstrates Improved Performance when Scaling up Image Resolution** We train `SOLO` on different image resolutions during the instruction fine-tuning stage for 1,000 steps due to the compute limits (see Fig. 9). The image resolution during inference matches that used in the instruction fine-tuning stage. We find that `SOLO` continues to improve the performance with increasing image resolution, especially for the visual mathematical reasoning task. In addition, there is no significant difference in performance between the 1024-square resolution and the adapted resolution with 1024 as the maximum used in `SOLO`. This demonstrates the efficiency and scalability of our flexible image pre-processing pipeline.

**`SOLO` Benefits from its Dynamic High-Resolution Capability** We evaluate LVLMs on high-resolution images and images with extreme aspect ratios that diverge from those found in natural images. For a controlled analysis, we select two benchmark datasets (ScienceQA, MathVista) where SOLO and

Table 5: Comparison of performance on ScienceQA and MathVista under conditions of extreme image resolutions.

| **Resolution > 800** | ScienceQA | MathVista | **Aspect Ratio > 3** | ScienceQA | MathVista |
|---|---|---|---|---|---|
| LLaVA-Next | 69.53 | 30.18 | LLaVA-Next | 43.28 | 25.00 |
| mLLAVA | 67.09 | 28.87 | mLLAVA | 34.45 | 21.88 |
| `SOLO` | **71.19** | **32.02** | `SOLO` | **50.42** | **28.13** |

LLaVA-Next achieve comparable performance (within 1 absolute point). In addition, we implement mLLaVA, which follows LLaVA's modeling framework but is trained using `SOLO`'s recipe for further comparison. We select subsets from the original benchmarks that include images with either a width or height exceeding 800 pixels. Also, we select those subsets with a ratio (width/height) greater than 3 or less than 1/3 for extreme aspect ratio analysis. We evaluate LVLMs' performance on these subsets. Our results, as shown in Tab. 5, verify `SOLO`'s advantages regarding the performance on high-resolution images and images with extreme aspect ratios. The poorer performance of mLLaVA and LLaVA-Next is likely due to their heuristic resizing rules, which constrain images to predefined resolution settings. In contrast, `SOLO` retains the original aspect ratios, which appears to be a more optimal approach.

# 7 Discussion

## 7.1 (Pre-)training Loss on Vision-Language Data is Not a Reliable Indicator of the Actual Performance

We find that both the pre-training loss and the instruction fine-tuning loss on the vision-language data are not reliable for the estimation of LVLMs' actual performance. References to support this claim regarding the pre-training loss include observations detailed in Fig. 3, Fig. 4, and Fig. 5a. Despite achieving similar

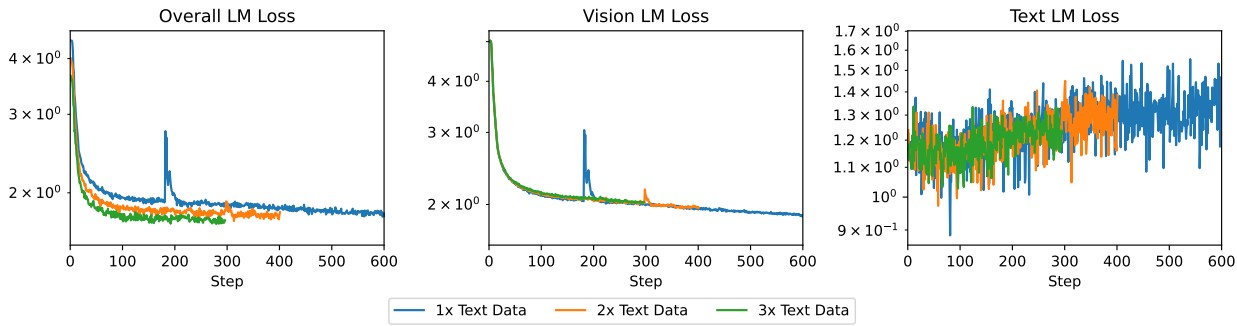

Figure 11: Stage 2 language modeling loss when trained on a mixture with different quantities of text data. 1x reflects the data mixture in Tab. 1, 2x and 3x represent mixtures with 2 or 3 times more text data compared to 1x while keeping the amount of vision data unchanged.

language modeling losses when conditioned on visual inputs, LVLMs exhibit markedly different behaviors and performance across various downstream tasks. This contrasts with findings from pure language modeling, where pre-training loss strongly correlates with various downstream task performance (Du et al., 2024).

We also demonstrate that the loss associated with the instruction fine-tuning data mixture does not reliably indicate task performance. We train a variant of SOLO with a learning rate of 1e-4, deviating from the prescribed rate of 1e-5 in our recipe. We show the training curves (Fig. 10a) and the downstream performance evaluation (Fig. 10b) of these two variants. The two variants exhibit similar training behaviors and losses on the instruction fine-tuning data mixture, yet they display significant performance disparities in downstream evaluation benchmarks.

Overall, our analysis highlights the need to identify a dependable metric for evaluating LVLMs with the unified architecture in pre-training, particularly for establishing scaling laws in future research.

## 7.2 Balancing Vision and Language Capabilities during Pre-Training is Challenging

We find that on a 7B scale, balancing vision and text capabilities can be challenging. Specifically, we observe that during Stage-2 pre-training, despite the inclusion of text-only pre-training data (§3.2.1) to maintain the language capability of the original LLM, the language modeling loss on the language-only pre-training subset still steadily increases as training continues. In Fig. 11, we introduce a setting where we gradually increase the proportion of text-only data per batch (Tab. 1) and monitor the language modeling loss for text. The results suggest that augmenting text data proportions does not alleviate the rise in language modeling loss, indicating challenges in achieving balanced vision and text capabilities in a 7B-scale model.

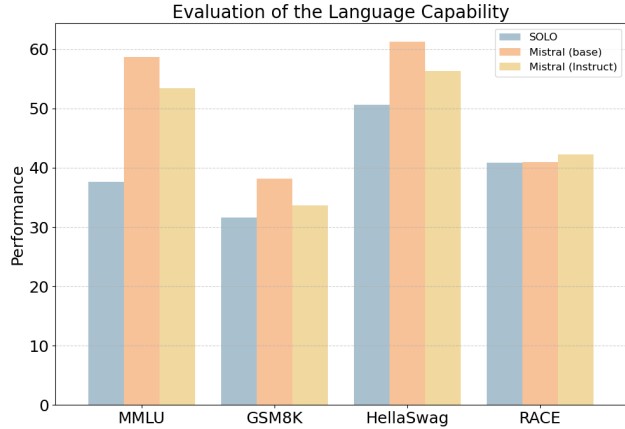

Figure 12: The evaluation of language capability.

To further understand the degradation in language ability of SOLO, we evaluate SOLO on standard LLM evaluation benchmarks, including MMLU (Hendrycks et al., 2020), GSM8k (Cobbe et al., 2021), HellaSwag (Zellers et al., 2019), and RACE (Lai et al., 2017). Our analysis includes comparisons with the backbone LLM of SOLO, specifically Mistral-7B-v0.1-base, as well as Mistral-7B-v0.1-Instruct (see Fig. 12). We observe a decline in language capabilities, particularly in knowledge-intensive benchmarks such as MMLU. There are two potential reasons: (1) Integrating vision capabilities may compromise language performance. (2) The quality of Mistral's pre-training corpus is better than the open-source Slimpajama we employ. Overall, the

Table 6: Comparison between `SOLO` and LVLMs with heterogeneous architectures regarding the complexity of scaling laws analysis.

| Comparison | SOLO | LVLMs with Heterogeneous Architecture |
|---|---|---|
| **Architecture** | Homogeneous | Heterogeneous (visual encoder, connector, LLM) |
| **Scaling laws analysis** | Standard approach (Hoffmann et al., 2022) | Requires extra "parameter allocation law" (an additional factor in the scaling law formulation) |
| **Key challenge** | N/A (widely explored in Hoffmann et al. (2022)) | Optimal parameter allocation and consistency issues |
| **Impact on computation** | Moderate | High, due to multiple configurations and additional experiments for parameter allocation law |
| **Risk of error propagation** | Low | Higher due to additional complexities |

current results indicate a limitation in the current version of `SOLO`, as effective performance in real-world vision-language tasks often necessitates strong foundational language capabilities, including knowledge and reasoning. Thus, we plan to maintain the language capabilities of `SOLO` in the upcoming version by enriching the pre-training dataset with a higher-quality text corpus and increasing the proportion of text data.

### 7.3   Scaling Laws Analysis with `SOLO`

We demonstrate how `SOLO` 's unified architecture facilitates easier analysis by comparing its experimental plan for deriving scaling laws to those of heterogeneous-architecture LVLMs. To simplify the analysis process, we follow the standard experimental setup in Hoffmann et al. (2022) to proportionally scale up the model size and training tokens by a factor of 20.

For `SOLO`, we can follow the standard scaling laws approach by selecting sampling models of varying sizes and assigning the appropriate number of training tokens based on the predefined scaling factor (*e.g.,* 20 times). After training each model on its designated tokens, the pre-training loss can thus be measured to obtain the (expended FLOPs, performance) data point. The obtained data points can be used to fit the power law curve that predicts the performance of the target model (*e.g.,* a model with 70B parameters) given the FLOPs expected to expend.

In contrast, for LVLMs with heterogeneous architectures, such as LLaVA, scaling laws analysis introduces additional complexity. Two key issues arise: (1) **Parameter allocation**: How should the model's parameters be distributed among the three components (visual encoder, connector, LLM)? (2) **Consistency during scaling**: Will this parameter allocation (*e.g.,* a 0.05:0.01:0.94 ratio) remain fixed as model size, data size, and compute scale up? To address these, an additional "parameter allocation law" must be derived to predict the optimal distribution of parameters based on model size and training tokens. In implementation, for each size of the sampling model, multiple configurations regarding parameter allocation must be tested to determine the optimal pre-training loss and the corresponding best allocation, significantly increasing computational demands. Moreover, it's challenging to figure out the best analytical form to map the model size and training tokens to the parameter allocation ratios. Consequently, the introduction of the parameter allocation law complicates analysis and increases the risk of error propagation.

## 8   Related Work

**Model Architecture**   Existing research advances the development of LVLMs capable of addressing diverse tasks via a unified interface that can directly generate natural language, thus avoiding task-specific modifications (Wang et al., 2021; 2022a; Li et al., 2023c). Utilizing pre-trained LLMs (Brown et al., 2020b; Bubeck et al., 2023) as the language component paired with pre-trained visual encoders (Radford et al., 2021; Dosovitskiy et al., 2021a), recent approaches further enhance the instruction-following, user-friendly responses generation, and complex reasoning ability of LVLMs (Liu et al., 2023c; Zhu et al., 2023; Dai et al., 2023; Alayrac et al., 2022; Li et al., 2023a; Ye et al., 2024). Concurrently, Wang et al. (2022b); Peng et al. (2022); Anil et al. (2023); Team (2024); Ge et al. (2023) propose to further learn a codebook in the initial stage to discretize the continuous embeddings extracted by visual encoders into a sequence of image tokens. These approaches enable a uniform vision-language modeling strategy for image and language tokens. However, the dependence on pre-trained visual encoders restricts the scalability of LVLMs. In this study, we address this challenge by readopting the conventional vision-language modeling approach that utilizes a single Transformer for both image and text processing (Li et al., 2019). Furthermore, while Bavishi et al. (2023)

extend this approach to billion-scale models, they do not disclose the specifics of their training processes. We address this gap by offering reproducible training recipes, complete with publicly released code, for scalable vision-language modeling on a 7-billion LVLM.

**Training Data**  Typically, LVLMs leverage extensive image-caption pair datasets (Lin et al., 2014a; Schuhmann et al., 2021; 2022; Yu et al., 2024; Chen et al., 2023b) to train a projector or a codebook that map continuous image features into the embedding space of LLMs, thereby aligning the two modalities (Li et al., 2023c; Gong et al., 2023; Zeng et al., 2023; Sun et al., 2023a). Furthermore, large-scale vision-language instruction tuning datasets (Su et al., 2023; Wei et al., 2023; Liu et al., 2023b; Gong et al., 2023; Gao et al., 2023; Li et al., 2023a) and feedback datasets (Li et al., 2023d; Sun et al., 2023b; Chen et al., 2024c; Zhang et al., 2024b) are utilized to further boost the fundamental capabilities of LVLMs and align LVLMs with human preferences, ensuring their ability to comprehend instructions and generate responses that are user-friendly. In this work, we propose a recipe that encompasses the selection of pre-training and instruction fine-tuning datasets, along with corresponding multi-stage paradigms, to facilitate the training of billion-scale LVLMs of a single Transformer architecture.

**Evaluation Benchmarks**  The progress of LVLMs is guided and measured by the continuous development of evaluation benchmarks (Ferraro et al., 2015; Kafle et al., 2019; Gan et al., 2022; Chen et al., 2024d). Initially, evaluation primarily concentrates on fundamental visual-language skills, such as image captioning (Lin et al., 2014b; Plummer et al., 2015), basic visual information recognition (Antol et al., 2015; Goyal et al., 2017), compositional visual understanding (Hudson & Manning, 2019), and knowledge reasoning based on visual information (Marino et al., 2019; Schwenk et al., 2022). Current benchmarks are advancing to encompass more intricate capabilities, requiring LVLMs to perform detailed visual analysis and complex reasoning (Uppal et al., 2022; Zhang et al., 2024a). These benchmarks range from general assessments across various domains and skills (Li et al., 2024a; Fu et al., 2024; Chen et al., 2024b; Yu et al., 2023) to specific tests targeting particular abilities, such as scientific document understanding (Kembhavi et al., 2016; Lu et al., 2022a), mathematical reasoning (Lu et al., 2023; Wang et al., 2024a), multi-discipline understanding and reasoning (Yue et al., 2023; Wu et al., 2024), hallucination (Li et al., 2023e), and OCR ability (Liu et al., 2023d). In this work, we select the advanced general and skill-specific benchmarks for evaluation.

## 9  Conclusion

This work revisits the simple vision-language modeling framework with a single Transformer. We argue that this approach effectively mitigates the scalability limitations inherent in prevailing models. With academic resources, we build `SOLO`, a 7B LVLM initialized from the Mistral LLM. We detail the training recipe and conduct extensive analysis and evaluation to validate the ingredients in our recipe. Experimental results show that `SOLO` demonstrates performance comparable to LLaVA-v1.5, supporting the continued investigation into this unified vision-language modeling approach for improved scalability.

## Limitations and Broader Impact Statement

The investigation into large-scale vision-language modeling using a unified transformer architecture remains nascent, with our model not yet reaching optimal performance across diverse benchmarks. Continued advancements in the direction of unified LVLMs for scalable vision-language modeling are anticipated. However, although developing LVLMs with strong capabilities brings significant advancements in AI, it also poses potential negative impacts. One concern is the risk of misuse, where the model could be employed for malicious purposes, such as generating misleading content that could manipulate public opinion or deceive individuals. Additionally, the model may inadvertently exacerbate biases present in the training data, leading to unfair or discriminatory outcomes in decision-making processes.

## Acknowledgement

We thank the reviewers and the action editor for their valuable suggestions and comments. This research is based upon work supported by U.S. DARPA ECOLE Program No. HR00112390060. The views and conclusions contained herein are those of the authors and should not be interpreted as necessarily representing the official policies, either expressed or implied, of DARPA, or the U.S. Government. The U.S. Government is authorized to reproduce and distribute reprints for governmental purposes notwithstanding any copyright annotation therein.

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

Table 7: Summary of datasets used in the supervised fine-tuning stage.

| Category | Dataset | #Sample |
|---|---|---|
| Language-Only | CodeAct-General (Wang et al., 2024b) | 71K |
| | UltraInteract-SFT (Yuan et al., 2024) | 288K |
| | UltraChat (Ding et al., 2023) | 207K |
| Detailed Image Caption | LVIS-Instruct4V (Wang et al., 2023a) | 223K |
| | ShareGPT4V (Chen et al., 2023b) | 102K |
| | LAION-GPT4V (LAION) | 12K |
| | Localized Narratives (Pont-Tuset et al., 2020) | 200K |
| | VSR (Liu et al., 2023a) | 2K |
| Scientific Document | TQA (Kembhavi et al., 2017) | 2K |
| | ScienceQA (Lu et al., 2022a) | 5K |
| Table, Document, and Chart | IconQA (Lu et al., 2021) | 27K |
| | TabMWP (Lu et al., 2022b) | 23K |
| | ChartQA (Masry et al., 2022) | 18K |
| | VisText (Tang et al., 2023) | 7K |
| | Chart2Text (Obeid & Hoque, 2020) | 27K |
| | DVQA (Kafle et al., 2018) | 20K |
| | FigureQA (Kahou et al., 2017) | 20K |
| OCR and Text-Rich Images | Diagram Image-to-Text (Kamizuru) | 300 |
| | Infographic VQA (Mathew et al., 2022) | 2K |
| | ST-VQA (Biten et al., 2019) | 17K |
| | TextCaps (Sidorov et al., 2020) | 22K |
| | TextVQA (Singh et al., 2019) | 22K |
| | OCR-VQA (Mishra et al., 2019) | 17K |
| | Rendered-Text (Wendler) | 10K |
| General VQA | HatefulMemes (Kiela et al., 2020) | 8.5K |
| | OK-VQA (Marino et al., 2019) | 9K |
| | AOK-VQA (Schwenk et al., 2022) | 16.5K |
| | TallyQA (Acharya et al., 2019) | 100K |
| | Visual7W (Zhu et al., 2016) | 14K |
| | COCO-QA (Ren et al., 2015) | 46K |
| | VQAV2 (Goyal et al., 2017) | 82K |
| | GQA (Hudson & Manning, 2019) | 72K |

# Appendix

# A  Details of Instruction-tuning Data Curation

The curated instruction fine-tuning data mixture is shown in Tab. 7. Each category is chosen to address specific challenges and capabilities of SOLO. For instance, datasets like UltraInteract-SFT (Yuan et al., 2024) and CodeAct-General (Wang et al., 2024b) enable the refinement of language processing and reasoning abilities, while visually rich datasets such as LVIS-Instruct4V (Wang et al., 2023a) and Localized Narratives (Pont-Tuset et al., 2020) enhance the model's basic image understanding and recognition abilities. Scientific document datasets like TQA (Kembhavi et al., 2017) are included to bolster the model's ability to parse and reason with academic visual information. Furthermore, OCR and text-heavy image datasets such as TextCaps (Sidorov et al., 2020) and OCR-VQA (Mishra et al., 2019) provide a crucial source for the model's ability to interpret text within complex images. By selecting datasets with a broad range of complexities, sizes, and focuses, we ensure a robust fine-tuning process that prepares SOLO to handle real-world applications effectively, reflecting a deep and detailed understanding of both vision and language data. Additionally, we conduct a thorough manual inspection and comparisons of the fine-tuning datasets, employing random sampling techniques on some datasets such as DVQA (Kafle et al., 2018) and FigureQA (Kahou et al., 2017) to guarantee diversity and prevent data imbalance.

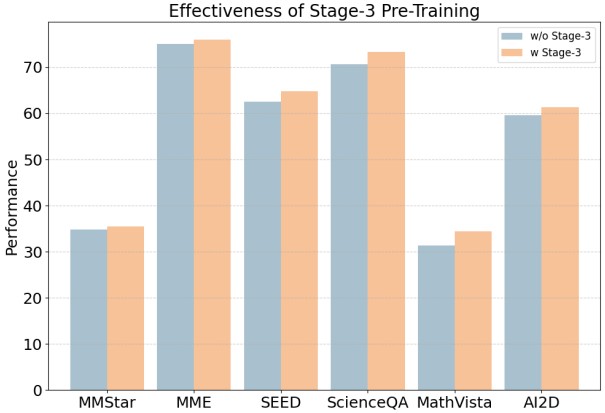 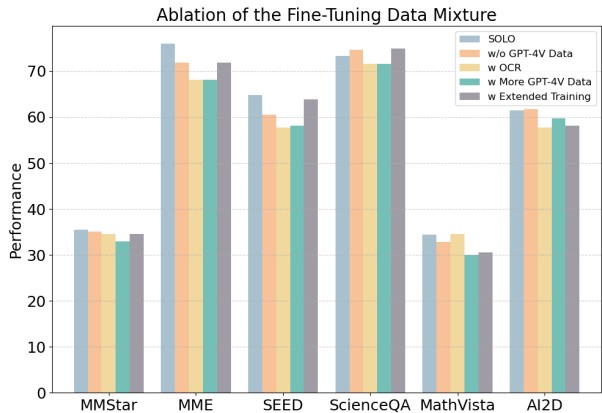

(a) The effectiveness of Stage-3 pre-training in priming SOLO for the instruction fine-tuning stage.

(b) The ablation study of the fine-tuning data mixture.

Figure 13: The evaluation performance of various ablations to validate the effectiveness of Stage-3 pre-training and the fine-tuning data mixture.

## B Stage-3 Annealing

We directly perform instruction fine-tuning on the pre-trained SOLO finished at Stage-2 to understand the effect of the annealing stage. The results shown in Fig. 13a indicate that introducing an annealing stage to conclude pre-training can slightly promote the performance across all evaluation benchmarks.

## C Effectiveness of Curated Data Mixture for Instruction Fine-Tuning

We conduct an ablation study to validate the curated data mixture for instruction fine-tuning. The ablations included are: (1) Without GPT-4V Data: All data generated by GPT-4V, including detailed captions and instructional fine-tuning samples, is excluded from the fine-tuning mixture. (2) With Additional OCR Data: Additional OCR data from LLaVAR is incorporated into the fine-tuning mixture to enhance OCR capabilities, which are crucial for tasks like require extract text information from charts. (3) With More GPT-4V Data: Data from GPT-4V used in the third stage of pre-training is added to the fine-tuning mixture. (4) Extended Training Duration: SOLO is trained for an additional epoch to investigate the effects of prolonged training.

The results are presented in Fig. 13b. Our analysis indicates that incorporating additional OCR data does not significantly enhance performance in scientific document comprehension or visual mathematical reasoning tasks, which rely extensively on visual text understanding. This lack of improvement can be attributed to the discrepancy between general OCR data and the specific demands of scientific charts. Identifying effective methods for collecting OCR data pertinent to scientific chart comprehension remains a critical area for future research. Furthermore, incorporating this OCR data seems to adversely affect overall visual-language capabilities, as demonstrated by general benchmarks. Regarding the use of GPT-4V data, our findings suggest that a measured inclusion during the pre-training annealing stage enhances performance (see §B), whereas excessive incorporation during fine-tuning can hurt the overall performance. We also find that SOLO exhibits minimal performance decline when trained solely on existing supervised datasets, excluding all data generated by GPT-4V. This demonstrates that GPT-4V data is not essential for enhancing the core capabilities of SOLO. The results overall justify the choice of datasets included in the supervised fine-tuning data mixture. However, although we observe a continual performance improvement across training steps within one epoch (see Fig. 5d), prolonged training on repetitive samples could lead to overfitting and decreased performance. This suggests that while extended exposure to diverse training data generally enhances model performance, overfitting remains a critical challenge when models are exposed repeatedly to a limited data subset. Overall, the ablation study proves the effectiveness of our curated data mixture.

