# OpenReview forum: "A Single Transformer for Scalable Vision-Language Modeling"
_TMLR — Accepted by TMLR_

### Review · Reviewer_Wo6i · 2024-08-26

**Summary Of Contributions:**

This paper presents a training recipe called Scalable Vision-Language Modeling (SOLO) for training a Large Vision-Language Model (LVLM), from ImageNet-scale pretraining to instructional tuning (fine-tuning). Unlike the LLaVA method, SOLO does not utilize an auxiliary vision encoder to project images into the LLM input embedding space. Instead, it directly maps image patches with a linear projection layer to allow image patches to be inputs of the LLM.

**Audience:**

No

**Broader Impact Concerns:**

Nothing special to be noticed in this paper. This paper proposes general LVLM training recipe, and would be good to consider impact of generative AI.

**Claims And Evidence:**

Yes

**Requested Changes:**

Please address the weaknesses to improve the paper. It would be beneficial to provide more analysis on the advantages of the proposed VisualBert-style architecture compared to LLaVA or other LVLM architectures.

**Strengths And Weaknesses:**

- Strength: The proposed method is simple and demonstrates performance that, while somewhat inferior, is comparable to existing LVLMs.

- Weakness: The main concern is the lack of novelty. The architectural contribution is limited, as the method closely resembles the VisualBert work. The authors themselves have stated that the proposed SOLO work is inspired by VisualBert. Additionally, I am concerned about the performance, as methods involving other vision encoders generally outperform the SOLO approach. Furthermore, the authors claim that heterogeneous architectures have limitations, such as difficulties with hardware infrastructure, but there is no empirical evidence or investigation of this issue presented in the paper.

---

> ### Author Response · Authors · 2024-09-13
>
> We sincerely thank the reviewer for the thorough evaluation and insightful comments on our manuscript. Below, we respond to the weaknesses respectively.
> # Weakness
> > The main concern is the lack of novelty. The architectural contribution is limited, as the method closely resembles the VisualBert work
>
> Although SOLO is inspired by VisalBERT, it is distinguished by significant and fundamental differences. We clarify some key distinctions as follows. VisualBERT relies on an object detector to identify bounding regions in images, which are then processed by a convolutional neural network to extract visual features. This also results in a model with a heterogeneous architecture. In contrast, SOLO doesn’t incorporate any external visual module (e.g., object detector and encoder for feature extraction), enabling a unified architecture that potentially minimizes the scalability concerns outlined in Sec.2. While SOLO is inspired by VisualBERT’s joint visual-text representation learning, their core strategies differ significantly.
>
> In addition, VisualBERT only successfully trains a model with 110 million parameters, and to the best of our knowledge, the unified modeling strategy has not been adopted in contemporary large-scale VLMs. This suggests that scaling VLMs with unified architectures to the billion-parameter scale introduces novel challenges. Our study is the first to articulate the critical benefits of unified architectures in the scalability of modern LVLMs and provide concrete evidence in Sec.6 to illustrate these scalability advantages. Also, we provide a detailed training recipe to successfully train such unified LVLMs with 7-billion parameters.
>
> We will clarify the distinctions and innovations in the revised version.
>
> > I am concerned about the performance
>
> We acknowledge that the performance of SOLO still lags behind the state-of-the-art (SOTA) LVLMs with heterogeneous architectures. However, it is important to consider that SOLO was trained using limited academic resources, specifically 8 A100 GPUs. This is in sharp contrast with the resources used to produce SOTA LVLMs. For example, in the technical report of DeepSeek-VL [1], the authors mentioned that DeepSeek-VL-7B was trained on a cluster of 64 nodes, each comprising 8 Nvidia A100 GPUs, totaling 512 GPUs—this represents a resource scale 64 times greater than that used for SOLO.
>
> SOLO serves as a pivotal model, showcasing the scientific value of training LVLMs with unified architectures to achieve enhanced scalability. Our research demonstrates that even with moderate academic resources, we can match or surpass the performance of other academically resourced LVLMs, such as LLaVA-v1.5 and MiniCPM-V.
>
> [1] DeepSeek-VL: Towards Real-World Vision-Language Understanding. Haoyu Lu et al
>
>
> > The authors claim that heterogeneous architectures have limitations, but there is no empirical
>
> Thanks for pointing it out! We conduct a detailed analysis to measure the inference speed of SOLO compared to several LVLMs with heterogeneous architectures, including LLaVA-Next, LLaVA-Interleave, and Qwen-VL. The evaluation is conducted on a single A100 GPU using 10,000 COCO images and a consistent prompt:  “Generate the Caption for the <image>”. All inference code is implemented using HuggingFace, and the inference latency is measured from the input being provided to the model and the output of the first token for fair comparison among all models since they may generate free-form responses in different lengths. The results, shown below, indicate that SOLO consistently outperforms other LVLMs with heterogeneous architectures by a significant margin, demonstrating its clear advantage in inference speed.
> | Model              | Inference Latency (second) |
> |--------------------|-------------------|
> | LLaVA-Next         | 0.0641            |
> | LLaVA-Interleave   | 0.0691            |
> | Qwen2-VL           | 0.0588            |
> | SOLO               | 0.0235           |
>
> In addition, we also measure the training speed of SOLO compared to LVLMs with heterogeneous architectures by measuring throughput. Using the same 8xA100 server, we compare the number of tokens processed per second during SOLO training and the official LLaVA implementation. Our SOLO implementation achieves a throughput of 20K tokens per second, while LLaVA reaches 10.5K tokens per second, highlighting SOLO's significant advantage in training speed.

---

> > ### Author Response · Authors · 2024-09-13
> >
> > # Requested Changes: More Analysis about the Advantages of SOLO compared to LLaVA
> > Thanks for pointing it out! We summarize the new discussion, analysis, and experiments we include during the author response period to further substantiate the advantages of SOLO compared to LVLMs with heterogeneous architectures like LLaVA.
> > 1. **SOLO facilitates easier scaling laws analysis**
> >
> > We demonstrate how SOLO’s unified architecture facilitates easier analysis by comparing its experimental plan for scaling laws to those of heterogeneous architecture LVLMs. To simplify the analysis process, we follow the standard experimental setup in [1] to proportionally scale up the model size and training tokens by a factor of 20.
> >
> > For SOLO, we can adhere to the standard scaling laws approach by selecting sampling models of varying sizes and assigning the appropriate number of training tokens based on the predefined scaling factor (i.e., 20 times). After training each model on its designated tokens, the pre-training loss can thus be measured to obtain the (expended FLOPs, performance) data point. The obtained data points can be used to fit the power law curve that predicts the performance of the target model (e.g., a model with 70B parameters) given the FLOPs expected to expend.
> >
> > In contrast, for LVLMs with heterogeneous architectures, such as LLaVA, scaling laws analysis introduces additional complexity. Two key issues arise: (1) **Parameter allocation**: How should the model’s parameters be distributed among the three components (visual encoder, connector, LLM)? (2) **Consistency during scaling**: Will this parameter allocation (e.g., a 0.05:0.01:0.94 ratio)  remain fixed as model size, data size, and compute scale up?
> >
> > To address these, a “parameter allocation law” must be derived to predict the optimal distribution of parameters based on model size and training tokens. In implementation, for each size of the sampling model, multiple configurations regarding parameter allocation must be tested to determine the optimal pre-training loss and the corresponding best allocation, significantly increasing computational demands. Moreover, it’s challenging to figure out the best analytical form to map the model size and training tokens to the parameter allocation ratios. Consequently, the introduction of the parameter allocation law complicates analysis and increases the risk of error propagation.
> >
> > We highlight key distinctions in the table below.
> >
> > | **Comparison**              | **SOLO**                                         | **LVLMs with Heterogenous Architecture**                        |
> > |-----------------------------|--------------------------------------------------|------------------------------------------------|
> > | **Architecture**             | Homogeneous                                     | Heterogeneous (visual encoder, connector, LLM) |
> > | **Scaling laws analysis**    | Standard approach [1]                               | Requires extra "parameter allocation law  (an additional factor in the scaling law formulation)"            |
> > | **Key challenge**            | N/A (widely explored in [1])             | Optimal parameter allocation and consistency issues    |
> > | **Impact on computation**    | Moderate                                        | High, due to multiple configurations and additional experiments for parameter allocation law           |
> > | **Risk of error propagation**| Low                                             | Higher due to additional complexities          |
> >
> > In addition, we conduct a simplified scaling laws experiment to show that the performance gain from scaling up data for SOLO is more predictable compared to LLaVA. Specifically, we follow [2] to fit the analytical function that models trained with a limited dataset (instruction fine-tuning in our case):
> >
> > $$
> > L(D) = \left( \frac{D_c}{D} \right)^{\alpha_D}
> > $$
> >
> >
> > We use data points from the instruction fine-tuning of both models to fit this function and report the coefficient of determination  $R^{2} $,  which reflects the quality of the fit and the predictability of performance. The results below demonstrate that SOLO’s performance is more predictable compared to LLaVA, indicating that SOLO facilitates easier scaling laws analysis.
> >
> > |       | MMStar | MME   | SEEDBENCH_II | ScienceQA | MathVista | AI2D   |
> > |-------|--------|-------|--------------|-----------|-----------|--------|
> > | SOLO  | 88.75  | 92.36 | 98.59        | 97.75     | 84.7      | 96.55  |
> > | mLLAVA| 40.32  | 92.52 | 92.89        | 89.25     | 62.73     | 83.29  |
> >
> >
> > [1] Training Compute-Optimal Large Language Models. Jordan Hoffmann et al
> >
> > [2] Scaling Laws for Neural Language Models. Jared Kaplan et al

---

> > > ### Author Response · Authors · 2024-09-13
> > >
> > > 2. **The inference latency of SOLO is smaller than LVLMs with heterogeneous architectures**
> > >
> > > Please refer to our responses to the third weakness.
> > >
> > >
> > > 3. **The advantages of SOLO’s dynamic high-resolution image preprocessing**
> > >
> > > We conduct additional experiments to explore the dynamic high-resolution capabilities of SOLO. Our results verify SOLO’s advantages regarding the performance on high-resolution images and images with weird aspect ratios. For a controlled analysis, we select two benchmark datasets (ScienceQA, MathVista) where SOLO and LLaVA-Next achieve comparable performance (within 1 absolute point). In addition, we implement mLLaVA, which follows LLaVA’s modeling framework but is trained using SOLO’s recipe for further comparison.
> > >
> > > **Evaluation of high-resolution image performance**: We select subsets from the original benchmarks that include images with either a width or height exceeding 800 pixels. We evaluate models’ performance on these subsets. Our analysis reveals that SOLO consistently surpasses both LLaVA-Next and mLLaVA in these high-resolution subsets, confirming the effectiveness of SOLO's high-resolution capabilities.
> > >
> > > **Dynamic image resolution analysis**: We examine subsets of images with extreme aspect ratios that diverge from those found in natural images. Specifically, we select those with a ratio (width/height) greater than 3 or less than 1/3. Our findings indicate that SOLO significantly outperforms LLaVA-Next and mLLaVA on these subsets. The poorer performance of mLLaVA and LLaVA-Next is likely due to their heuristic resizing rules, which constrain images to predefined resolution settings. In contrast, SOLO retains the original aspect ratios, which appears to be a more optimal approach.
> > >
> > >
> > > | Resolution > 800 | ScienceQA | MathVista | Aspect Ratio > 3 | ScienceQA | MathVista |
> > > |------------------|-----------|-----------|------------------|-----------|-----------|
> > > | LLaVA-Next       | 69.53     | 30.18     | LLaVA-Next       | 43.28     | 25.00        |
> > > | mLLAVA           | 67.09     | 28.87     | mLLAVA           | 34.45     | 21.88     |
> > > | SOLO             | 71.19     | 32.02     | SOLO             | 50.42     | 28.13     |
> > >
> > > In addition, we also have one dedicated section 6 in our paper to verify the better scalability of SOLO compared to LLaVA. We would like to highlight the results described in Figure 8. We conducted a controlled analysis during the instruction fine-tuning phase, comparing SOLO to our mLLaVA model, which follows the same training recipe. Since both models were fine-tuned on the same dataset, we measured and compared their average performance improvement per token on the target benchmarks.
> > >
> > > As shown in Figure 8, SOLO consistently outperforms mLLaVA on this "performance improvement per token" metric across all evaluation benchmarks. This suggests that SOLO benefits more from high-quality instruction fine-tuning data and has better scalability, with the potential for further improvements as the dataset size increases.

---

> > > > ### Author Response · Authors · 2024-09-18
> > > >
> > > > Hi Reviewer,
> > > >
> > > > Please accept our sincere gratitude for all your suggestions on our work. we hope our responses have answered your questions. We are happy to discuss any questions you may have at this stage!
> > > >
> > > > Best, Authors

---

### Review · Reviewer_4ucY · 2024-08-30

**Summary Of Contributions:**

This paper introduces SOLO, a novel approach to vision-language modeling that employs a single Transformer architecture for both image and text processing. The key contributions are:
- A unified architecture that potentially addresses scalability limitations in current large vision-language models (LVLMs) that use separate visual encoders and language models.
- A comprehensive training recipe, including a three-stage pre-training process and instruction-supervised fine-tuning, implemented using moderate academic resources (8 x A100 80GB GPUs).
- Empirical evidence showing SOLO's performance is comparable to state-of-the-art LVLMs across various benchmarks, with particular excellence in visual mathematical reasoning tasks.
- Detailed ablation studies validating the effectiveness of each component in the training recipe.
- Analysis of SOLO's scaling behavior, demonstrating improved performance with increasing image resolution.

These contributions are likely to be of significant interest to TMLR's audience, particularly those working in the fields of vision-language modeling, multimodal AI, and large language models.

**Audience:**

Yes

**Broader Impact Concerns:**

The authors have adequately addressed potential negative impacts of LVLM development in their Limitations and Broader Impact Statement. They demonstrate awareness of the risks of misuse and the potential for bias amplification. This section concisely covers key ethical considerations, which appears appropriate for the scope of the paper.

**Claims And Evidence:**

Yes

**Requested Changes:**

Critical changes:

1. Provide a more comprehensive analysis of scaling laws. This should include:
a) A clearer demonstration of how SOLO's architecture facilitates easier derivation of scaling laws compared to other LVLMs.
b) Discussion on how scaling laws might differ when considering data modalities separately (image-only, image-text, text-only) versus collectively.
2. Include inference speed comparisons with other LVLMs to substantiate the potential advantages of the unified architecture.
3. Expand on the rationale behind using different training frameworks (Megatron and Accelerate) for different stages. This should include:
a) A discussion of the potential difficulties practitioners might face when using Megatron.
b) Quantitative data on the speed advantages of Megatron over Accelerate during pre-training.

Suggested improvements:

1. Provide a more comprehensive analysis of SOLO's performance with dynamic aspect ratios and very high-resolution images. This should include a direct comparison with models like LLaVA-Next, which have shown significant performance gains through advanced high-resolution techniques. Such an analysis would help clarify whether SOLO is fully leveraging its flexible resolution capabilities to maximize performance.
2. Explore the impact of intermediate datasets (e.g., OpenImages and its comprehensive label set) for stage 1 pre-training as a bridge between single-label datasets like ImageNet and noisy web-crawled captions. Analyze how incorporating such datasets might affect model performance and understanding of complex visual concepts.
3. Investigate and report on how the ratio of image to text tokens during pre-training affects the model's performance. This analysis would provide valuable insights into the optimal balance between visual and linguistic inputs for large-scale vision-language models, particularly at the 7B parameter scale.

**Strengths And Weaknesses:**

Strengths:

- Novel unified architecture: The single Transformer approach for both vision and language processing is innovative and addresses several scalability issues in current LVLMs.
- Comprehensive training recipe: The detailed three-stage pre-training process and instruction fine-tuning provide a valuable blueprint for future open research.
- Extensive ablation studies: The paper includes thorough analyses that validate the key components of their approach.
- Strong performance: SOLO achieves comparable or superior performance to existing LVLMs, particularly in visual mathematical reasoning tasks.
- Scalability analysis: The paper provides some insights into the scaling behavior of SOLO. However, this analysis could be more comprehensive and comparative with other LVLM architectures.

Weaknesses:

- Insufficient exploration of dynamic high-resolution capabilities: While SOLO supports flexible image resolutions, the paper lacks a thorough analysis of its performance with dynamic aspect ratios and very high-resolution images. This is particularly notable when compared to models like LLaVA-Next, which uses the same Mistral-7B base but achieves better performance, largely attributed to its dynamic high-resolution technique. The paper doesn't adequately address whether SOLO fully leverages its potential to process images with dynamic aspect ratios and high resolutions to boost performance. A more in-depth analysis and comparison in this aspect would have been valuable, especially given that this capability is a key factor in the superior performance of some competing models.
- Incomplete analysis of scaling laws: The paper mentions the importance of reliable scaling laws early on, but does not sufficiently demonstrate how SOLO's unified architecture facilitates easier derivation of scaling laws compared to other LVLMs. Additionally, the paper lacks discussion on how scaling laws might differ when considering data modalities separately (image-only, image-text, text-only) versus collectively.
- Lack of inference speed comparison: Despite the potential speed advantages of the unified architecture, the paper doesn't provide comparisons of inference speed with other models, particularly more recent LVLMs.
- Insufficient explanation of framework choices: The paper doesn't adequately explain the rationale behind using different training frameworks (Megatron and Accelerate) for different stages, nor does it quantify the benefits of these choices.
- Limited exploration of intermediate datasets: The transition from single-label datasets like ImageNet to noisy web-crawled captions isn't explored in depth, missing potential insights from intermediate datasets.

---

> ### Author Response · Authors · 2024-09-13
>
> We sincerely thank the reviewer for the very detailed comments and suggestions. Below, we respond to the weaknesses respectively.
> # Requested changes
> > Provide a more comprehensive analysis of scaling laws.
>
> Thanks for pointing it out! We address the two comments respectively.
>
> **How SOLO's architecture facilitates easier derivation**: We demonstrate how SOLO’s unified architecture facilitates easier analysis by comparing its experimental plan for scaling laws to those of heterogeneous-architecture LVLMs. To simplify the analysis process, we follow the standard experimental setup in [1] to proportionally scale up the model size and training tokens by a factor of 20.
>
> For SOLO, we can adhere to the standard scaling laws approach by selecting sampling models of varying sizes and assigning the appropriate number of training tokens based on the predefined scaling factor (i.e., 20 times). After training each model on its designated tokens, the pre-training loss can thus be measured to obtain the (expended FLOPs, performance) data point. The obtained data points can be used to fit the power law curve that predicts the performance of the target model (e.g., a model with 70B parameters) given the FLOPs expected to expend.
>
> In contrast, for LVLMs with heterogeneous architectures, such as LLaVA, scaling laws analysis introduces additional complexity. Two key issues arise: (1) **Parameter allocation**: How should the model’s parameters be distributed among the three components (visual encoder, connector, LLM)? (2) **Consistency during scaling**: Will this parameter allocation (e.g., a 0.05:0.01:0.94 ratio)  remain fixed as model size, data size, and compute scale up?
> To address these, a “parameter allocation law” must be derived to predict the optimal distribution of parameters based on model size and training tokens. In implementation, for each size of the sampling model, multiple configurations regarding parameter allocation must be tested to determine the optimal pre-training loss and the corresponding best allocation, significantly increasing computational demands. Moreover, it’s challenging to figure out the best analytical form to map the model size and training tokens to the parameter allocation ratios. Consequently, the introduction of the parameter allocation law complicates analysis and increases the risk of error propagation.
>
> We highlight key distinctions in the table below.
>
> | **Comparison**              | **SOLO**                                         | **LVLMs with Heterogenous Architecture**                        |
> |-----------------------------|--------------------------------------------------|------------------------------------------------|
> | **Architecture**             | Homogeneous                                     | Heterogeneous (visual encoder, connector, LLM) |
> | **Scaling laws analysis**    | Standard approach [1]                               | Requires extra "parameter allocation law  (an additional factor in the scaling law formulation)"            |
> | **Key challenge**            | N/A (widely explored in [1])             | Optimal parameter allocation and consistency issues    |
> | **Impact on computation**    | Moderate                                        | High, due to multiple configurations and additional experiments for parameter allocation law           |
> | **Risk of error propagation**| Low                                             | Higher due to additional complexities          |
>
>
> In addition, we conduct a simplified scaling laws experiment to show that the performance gain from scaling up data for SOLO is more predictable compared to LLaVA. Specifically, we follow [2] to fit the analytical function that models trained with a limited dataset (instruction fine-tuning in our case):
>
> $$
> L(D) = \left( \frac{D_c}{D} \right)^{\alpha_D}
> $$
>
>
> We use data points from the instruction fine-tuning of both models to fit this function and report the coefficient of determination  $ R^{2} $,  which reflects the quality of the fit and the predictability of performance. The results below demonstrate that SOLO’s performance is more predictable compared to LLaVA, indicating that SOLO facilitates easier scaling laws analysis.
>
> |       | MMStar | MME   | SEEDBENCH_II | ScienceQA | MathVista | AI2D   |
> |-------|--------|-------|--------------|-----------|-----------|--------|
> | SOLO  | 88.75  | 92.36 | 98.59        | 97.75     | 84.7      | 96.55  |
> | mLLAVA| 40.32  | 92.52 | 92.89        | 89.25     | 62.73     | 83.29  |
>
>
> [1] Training Compute-Optimal Large Language Models. Jordan Hoffmann et al
>
> [2] Scaling Laws for Neural Language Models. Jared Kaplan et al

---

> > ### Author Response · Authors · 2024-09-13
> >
> > **How scaling laws might differ when considering data modalities separately versus collectively**: Considering data modalities separately, as in vision-language scaling laws analysis, essentially introduces several new constant terms in the analytical form that need to be estimated to address the need of adjusting mixing data ratios of different modalities (e.g., image-only, text-only). In implementation, this involves partitioning the total expended FLOPs into modality-specific FLOPs, experimenting with different combinations of mixing ratios, and measuring their performance to obtain data points, which can be used to fit the curve that maps 3 modality-specific FLOPs to the final performance. This closely parallels the analysis of how mixing data from different domains affects LLM pre-training, as systematically examined in [1].
> >
> > [1] Data Mixing Laws: Optimizing Data Mixtures by Predicting Language Modeling Performance; Jiasheng Ye et al
> >
> >
> > > Include inference speed comparisons with other LVLMs
> >
> > Thanks for pointing it out! We conduct a detailed analysis to measure the inference speed of SOLO compared to several LVLMs with heterogeneous architectures, including LLaVA-Next, LLaVA-Interleave, and Qwen-VL. The evaluation is conducted on a single A100 GPU using 10,000 COCO images and a consistent prompt:  “Generate the Caption for the <image>”. All inference code is implemented using HuggingFace, and the inference latency is measured from from the input being provided to the model and the output of the first token for fair comparison among all models since they may generate free-form responses in different lengths. The results, shown below, indicate that SOLO consistently outperforms other LVLMs with heterogeneous architectures by a significant margin, demonstrating its clear advantage in inference speed.
> > | Model              | Inference Latency (second) |
> > |--------------------|-------------------|
> > | LLaVA-Next         | 0.0641            |
> > | LLaVA-Interleave   | 0.0691            |
> > | Qwen2-VL           | 0.0588            |
> > | SOLO               | 0.0235           |
> >
> > In addition, we also measure the training speed of SOLO compared to LVLMs with heterogeneous architectures by measuring throughput. Using the same 8xA100 server, we compare the number of tokens processed per second during SOLO training and the official LLaVA implementation. Our SOLO implementation achieves a throughput of 20K tokens per second, while LLaVA reaches 10.5K tokens per second, highlighting SOLO's significant advantage in training speed.
> >
> >
> >
> > > Expand on the rationale behind using different training frameworks for different stages.
> >
> > Thanks for pointing it out! We address the two comments respectively.
> >
> > **Potential difficulties when using Megatron**: The decision to use Accelerate DeepSpeed over Megatron for post-training (i.e., instruction fine-tuning) is driven by practical considerations. Accelerate allows for faster experiments with different data mixtures by changing a few lines of code, while Megatron requires extensive preprocessing (pre-tokenizing and saving them to files) for each run of different data mixtures, which can also complicate ablation studies.  In addition, Megatron’s complex implementation, with multiple layers of packaging, limits the ease of customization—especially critical when introducing advanced methods like RLHF to improve post-training. Thus, Accelerate offers greater flexibility and extensibility in the post-training phase.
> >
> > **Quantitative data on the speed advantages of Megatron over Accelerate**: We conduct a controlled analysis to compare the training throughput of the Megatron and Accelerate DeepSpeed frameworks. Using identical image-caption pre-training on an 8xA100 server, we measure the number of training tokens processed per second by each framework. Megatron achieves a throughput of 20K tokens per second, while Accelerate reaches 12K tokens per second. The results indicate that Megatron demonstrates a significantly higher training throughput compared to Accelerate, processing nearly 67% more tokens per second under identical conditions. This suggests that for tasks requiring high training efficiency, Megatron may offer a more optimal solution. In our case of pre-training, given the longer pre-training duration and lack of necessity for an ablation study on pre-training datasets, Megatron is the more suitable framework for our needs.

---

> > > ### Author Response · Authors · 2024-09-13
> > >
> > > > Insufficient exploration of dynamic high-resolution capabilities
> > >
> > > Thanks for pointing it out! We conduct additional experiments to explore the dynamic high-resolution capabilities of SOLO. Our results verify SOLO’s advantages regarding the performance on high-resolution images and images with weird aspect ratios. For a controlled analysis, we select two benchmark datasets (ScienceQA, MathVista) where SOLO and LLaVA-Next achieve comparable performance (within 1 absolute point). In addition, we implement mLLaVA, which follows LLaVA’s modeling framework but is trained using SOLO’s recipe for further comparison.
> > >
> > > **Evaluation of high-resolution image performance**: We select subsets from the original benchmarks that include images with either a width or height exceeding 800 pixels. We evaluate models’ performance on these subsets. Our analysis reveals that SOLO consistently surpasses both LLaVA-Next and mLLaVA in these high-resolution subsets, confirming the effectiveness of SOLO's high-resolution capabilities.
> > >
> > > **Dynamic image resolution analysis**: We examine subsets of images with extreme aspect ratios that diverge from those found in natural images. Specifically, we select those with a ratio (width/height) greater than 3 or less than 1/3. Our findings indicate that SOLO significantly outperforms LLaVA-Next and mLLaVA on these subsets. The poorer performance of mLLaVA and LLaVA-Next is likely due to their heuristic resizing rules, which constrain images to predefined resolution settings. In contrast, SOLO retains the original aspect ratios, which appears to be a more optimal approach.
> > >
> > >
> > > | Resolution > 800 | ScienceQA | MathVista | Aspect Ratio > 3 | ScienceQA | MathVista |
> > > |------------------|-----------|-----------|------------------|-----------|-----------|
> > > | LLaVA-Next       | 69.53     | 30.18     | LLaVA-Next       | 43.28     | 25.00        |
> > > | mLLAVA           | 67.09     | 28.87     | mLLAVA           | 34.45     | 21.88     |
> > > | SOLO             | 71.19     | 32.02     | SOLO             | 50.42     | 28.13     |
> > >
> > >
> > >
> > > > Explore the impact of intermediate datasets
> > >
> > > Thanks for pointing it out! We conduct a controlled analysis to understand the impact of intermediate datasets. Due to the computational constraints, we randomly select 2M examples from OpenImages, concatenating all image labels with commas to form captions for pre-training. Following this, the model is further pre-trained on web-scale data for 150 steps and is instruction fine-tuned for 1,000 steps to ensure comparability with our existing checkpoint. The results are shown below. We do not observe significant performance changes regarding our evaluation benchmarks when introducing the OpenImage dataset for intermediate pre-training. This is likely because ImageNet-21K already encompasses a wide range of visual concepts and the prolonged pre-training of SOLO on this dataset, rendering further pre-training on concept recognition less impactful.
> > >
> > >
> > > |                      | MMStar | MME   | SEED | ScienceQA | MathVista | AI2D  |
> > > |----------------------|--------|-------|------|-----------|-----------|-------|
> > > | Stage-1+Stage-2       | 29.33  | 979   | 44.47| 67.5      | 28        | 48.31 |
> > > | Stage-1+OpenImages+Stage-2 | 28.45  | 988   | 45.36| 68.3      | 27.6      | 46.82 |
> > >
> > > > Investigate and report on how the ratio of image to text tokens during pre-training affects the model's performance.
> > >
> > > Thanks for the suggestion. We do have one dedicated section in Appendix-F that presents a detailed analysis about the ratio of image-to-text tokens. We introduce a setting where we gradually increase the proportion of text-only data per batch (1x, 2x, 3x) and monitor the language modeling loss for text (Figure 12). The results suggest that augmenting text data proportions does not alleviate the rise in language modeling loss, indicating challenges in achieving balanced vision and text capabilities in a 7B-scale model. There are two potential reasons: (1) Integrating vision capabilities may compromise language performance. (2) The quality of Mistral’s pre-training corpus is better than the open-source Slimpajama we employ. Overall, the current results indicate a limitation in the current version of SOLO, as effective performance in real-world vision-language tasks often necessitates strong foundational language capabilities, including knowledge and reasoning. Thus, we plan to maintain the language capabilities of SOLO in the upcoming version by enriching the pre-training dataset with a higher-quality text corpus and increasing the proportion of text data.

---

> > > > ### Author Response · Authors · 2024-09-18
> > > >
> > > > Hi Reviewer,
> > > >
> > > > Please accept our sincere gratitude for all your suggestions on our work. we hope our responses have answered your questions. We are happy to discuss any questions you may have at this stage!
> > > >
> > > > Best, Authors

---

### Review · Reviewer_Fr5d · 2024-09-06

**Summary Of Contributions:**

The authors present the Unified End-to-End Transformer, dubbed SOLO, mitigating four scalability issues they identified: limited visual capabilities, difficulties in large-scale training and deployment, complexity in scaling analysis due to multiple components, and restricted flexibility in image pre-processing. Their model is fully open-source, accompanied by a detailed training recipe that includes initialization, sequential pre-training on diverse datasets, and instruction fine-tuning. Through extensive evaluation, SOLO demonstrates performance comparable to LLaVA-v1.5-7B, with notable strengths in visual mathematical reasoning.

**Audience:**

Yes

**Broader Impact Concerns:**

No additional concerns on the ethical implications of this work

**Claims And Evidence:**

Yes

**Requested Changes:**

All requested changes are described in the Weaknesses section

**Strengths And Weaknesses:**

### Strengths

1. **Strong Empirical Results with Unified VLM Architecture**: the proposed model architecture (similar to early models like VisualBERT and PixelBERT) shows strong performance, particularly outperforming in Level-1 and Level-2 categories.

2. **Thorough Experimental Analysis**: The authors conduct comprehensive experiments to validate SOLO's effectiveness across various benchmark datasets. They also provide detailed insights into the training process, highlighting key factors such as the importance of different stages in sequential pre-training, performance improvements with increased token length, and gains from higher image resolution.


### Weaknesses

1. **Ambiguity in the Notion of "Scalability"**: The paper’s central claim is the "scalability" of LVLM, but the definition of scalability is somewhat vague. Typically, scalability refers to expanding the model architecture or dataset size. However, in this paper, scalability focuses on visual capacity, token length, and image resolution. The scope of scalability should be clarified to ensure readers fully understand which aspects are being scaled.
   -  Based on my understanding, the authors also argue that the unified architecture could offer greater benefits if it becomes scalable in terms of model size and dataset in the future. However, to support this claim, additional experiments and analyses focused on scaling the model size and dataset should be actually included. Without such experiments, the claim of future scalability remains speculative.

2. **Unclear Motivation and Goal (Position)**:  While the authors identify four limitations, the need for a unified single transformer remains unclear.

  -  For  1) Fixed and Constrained Visual Capabilities and 4) Limited Image Pre-Processing Flexibility:  the limitations seem to stem from using pre-trained visual encoders (e.g., CLIP ViT). The difference between previous approaches and this work appears to be based on two factors: 1) the use of a pre-trained visual encoder or training from scratch and 2) the depth of the visual extractor (deep vs. shallow). A more detailed analysis of these two factors is needed.
     - Moreover,  the authors suggest that increasing visual capacity can be achieved by training a shallow visual encoder from scratch in stage-1, without initializing from a pre-trained encoder. However, it’s unclear whether this represents a genuine contribution, or if a shallow encoder is truly better than a deep one.
     - Couldn’t a deep visual encoder (with or without initialization) also be trained in stage-1? What is the specific advantage of the unified model over these alternatives?
     - Additional analyses on performance improvements relative to computational costs, particularly regarding these two factors, are necessary.

 - For 3) Multiple Components Complicate the Scaling Analysis: the rationale for why a unified model simplifies scaling analysis is unclear. In fact, with previous models, using multiple components allows for a more isolated evaluation of each modality’s encoders. In contrast, the unified model appears more difficult to analyze since the effects of all components are intertwined, making it harder to assess each modality’s individual contribution.
 - For 2) Challenges in Large-Scale Training and Deployment: If possible, more discrete and real examples would be helpful as to why the model with multiple components is hard to train and deploy.

Given the above points and the potentially heavy computational overhead (my guess), it is uncertain whether the proposed model and training recipe offer significant practical benefits.


3. **Insufficient explanation in FIgure 8**: The performance improvements per token depicted in Figure 8 require a more detailed explanation of the setting and experiments. It’s unclear why SOLO shows greater improvement than LLaVA, and what this result means in terms of the model's overall capability.

4. **Limitations should be described**: the paper lacks a discussion on the limitations of the proposed model. From my understanding, SOLO likely incurs higher computational costs compared to previous methods. Additionally, a comparison of model parameter size with LLaVA would be useful for understanding the trade-offs between the two models.

---

> ### Author Response · Authors · 2024-09-13
>
> We sincerely thank the reviewer for posting insightful comments on our work. Below, we respond to the weaknesses respectively.
>
> # Weakness
> > Ambiguity in the Notion of "Scalability"
>
> Thanks for pointing it out! The scaling process does involve increasing the model size, training tokens, and expended compute. However, the “scalability” of an ML model, which we explicitly define in the second paragraph of the introduction, measures whether an architecture can demonstrate sustained performance improvement through scaling (e.g., more computational resources and/or data) without hitting any inherent bottleneck in the machine learning system.
>
> We agree that additional experiments on scaling model size and datasets would strengthen the scalability claim. However, due to limited computational resources in academia, this remains a limitation of our work. To address this, we carefully examine the potential inherent bottleneck in current heterogeneous-architecture LVLMs that can hinder scalability, including visual capacity, image resolution, and so on. We conduct a thorough analysis, ablate each potential bottleneck, and verify the claims made in the paper. Our extensive analysis in Section 6 and in response to weakness 2 highlight these factors as possible limitations in scaling, underscoring the advantages of the unified architecture used in SOLO.
>
> We will clarify this in the revision.

---

> > ### Author Response · Authors · 2024-09-13
> >
> > > Unclear Motivation and Goal (Position)
> >
> > Thanks for pointing it out! We address each point separately as follows.
> >
> > **Fixed and constrained visual capabilities and limited image pre-processing flexibility**: We concur that the key distinction between earlier LVLMs and SOLO lies in the use of a pre-trained visual encoder versus training from scratch. However, we would like to clarify that although we adopt a shallow visual extractor (i.e., a linear projector to convert raw image pixels into visual embeddings), we pre-train the whole model (the linear projector and the whole Transformer) for image representation learning in stage-1, instead of only training a shallow visual encoder. In this case, this unified architecture leverages the full 7B parameters of the Transformer for image representation learning, rather than being limited by a 300M-parameter pre-trained visual encoder. As a result, this approach essentially trains a deeper visual encoder from scratch, enhancing visual capability.
> >
> > If we use an extra visual encoder, it will  compromise the unified modeling approach employed in SOLO, leading to a heterogeneous architecture. We discuss the scalability limitations of such heterogeneous architectures in detail in Section 2 of our paper.
> >
> > We summarize our experiments that support the claims regarding fixed and constrained visual capabilities and limited image pre-processing flexibility as follows.
> >
> > **Fixed and constrained visual capabilities**: Please refer to our response to weakness 3 regarding additional explanation in Figure 8 (Section 6.2). In summary, the results suggest SOLO benefits more from high-quality instruction fine-tuning data and demonstrates better scalability, indicating that its performance could further be improved more compared to LLaVA-Style LVLMs with more data. This provides evidence that by unlocking the model's visual capabilities—specifically replacing the pre-trained 300M visual encoders and utilizing the full 7B parameters for image representation learning—the model's scalability can be significantly improved.
> >
> > **Limited Image Pre-Processing Flexibility**: We conduct additional experiments to explore the dynamic high-resolution capabilities of SOLO. Our results verify SOLO’s advantages regarding the performance on high-resolution images and images with weird aspect ratios. For a controlled analysis, we select two benchmark datasets (ScienceQA, MathVista) where SOLO and LLaVA-Next achieve comparable performance (within 1 absolute point). In addition, we implement mLLaVA, which follows LLaVA’s modeling framework but is trained using SOLO’s recipe for further comparison.
> >
> > Evaluation of high-resolution image performance: We select subsets from the original benchmarks that include images with either a width or height exceeding 800 pixels. We evaluate models’ performance on these subsets. Our analysis reveals that SOLO consistently surpasses both LLaVA-Next and mLLaVA in these high-resolution subsets, confirming the effectiveness of SOLO's high-resolution capabilities.
> >
> > Dynamic image resolution analysis: We examine subsets of images with extreme aspect ratios that diverge from those found in natural images. Specifically, we select those with a ratio (width/height) greater than 3 or less than 1/3. Our findings indicate that SOLO significantly outperforms LLaVA-Next and mLLaVA on these subsets. The poorer performance of mLLaVA and LLaVA-Next is likely due to their heuristic resizing rules, which constrain images to predefined resolution settings. In contrast, SOLO retains the original aspect ratios, which appears to be a more optimal approach.
> >
> > | Resolution > 800 | ScienceQA | MathVista | Aspect Ratio > 3 | ScienceQA | MathVista |
> > |------------------|-----------|-----------|------------------|-----------|-----------|
> > | LLaVA-Next       | 69.53     | 30.18     | LLaVA-Next       | 43.28     | 25.00        |
> > | mLLAVA           | 67.09     | 28.87     | mLLAVA           | 34.45     | 21.88     |
> > | SOLO             | 71.19     | 32.02     | SOLO             | 50.42     | 28.13     |

---

> ### Author Response · Authors · 2024-09-13
>
> **Multiple components complicate the scaling analysis**: We demonstrate how SOLO’s unified architecture facilitates easier analysis by comparing its experimental plan for scaling laws to those of heterogeneous architecture LVLMs. To simplify the analysis process, we follow the standard experimental setup in [1] to proportionally scale up the model size and training tokens by a factor of 20.
>
> For SOLO, we can adhere to the standard scaling laws approach by selecting sampling models of varying sizes and assigning the appropriate number of training tokens based on the predefined scaling factor (i.e., 20 times). After training each model on its designated tokens, the pre-training loss can thus be measured to obtain the (expended FLOPs, performance) data point. The obtained data points can be used to fit the power law curve that predicts the performance of the target model (e.g., a model with 70B parameters) given the FLOPs expected to expend.
>
> In contrast, for LVLMs with heterogeneous architectures, such as LLaVA, scaling laws analysis introduces additional complexity. Two key issues arise: (1) **Parameter allocation**: How should the model’s parameters be distributed among the three components (visual encoder, connector, LLM)? (2) **Consistency during scaling**: Will this parameter allocation (e.g., a 0.05:0.01:0.94 ratio)  remain fixed as model size, data size, and compute scale up?
>
> To address these, a “parameter allocation law” must be derived to predict the optimal distribution of parameters based on model size and training tokens. In implementation, for each size of the sampling model, multiple configurations regarding parameter allocation must be tested to determine the optimal pre-training loss and the corresponding best allocation, significantly increasing computational demands. Moreover, it’s challenging to figure out the best analytical form to map the model size and training tokens to the parameter allocation ratios. Consequently, the introduction of the parameter allocation law complicates analysis and increases the risk of error propagation.
> We highlight key distinctions in the table below.
> | **Comparison**              | **SOLO**                                         | **LVLMs with Heterogenous Architecture**                        |
> |-----------------------------|--------------------------------------------------|------------------------------------------------|
> | **Architecture**             | Homogeneous                                     | Heterogeneous (visual encoder, connector, LLM) |
> | **Scaling laws analysis**    | Standard approach [1]                               | Requires extra "parameter allocation law  (an additional factor in the scaling law formulation)"            |
> | **Key challenge**            | N/A (widely explored in [1])             | Optimal parameter allocation and consistency issues    |
> | **Impact on computation**    | Moderate                                        | High, due to multiple configurations and additional experiments for parameter allocation law           |
> | **Risk of error propagation**| Low                                             | Higher due to additional complexities          |
>
> In addition, we conduct a simplified scaling laws experiment to show that the performance gain from scaling up data for SOLO is more predictable compared to LLaVA. Specifically, we follow [2] to fit the analytical function that models trained with a limited dataset (instruction fine-tuning in our case):
>
> $$
> L(D) = \left( \frac{D_c}{D} \right)^{\alpha_D}
> $$
>
>
> We use data points from the instruction fine-tuning of both models to fit this function and report the coefficient of determination  $ R^{2}$,  which reflects the quality of the fit and the predictability of performance. The results below demonstrate that SOLO’s performance is more predictable compared to LLaVA, indicating that SOLO facilitates easier scaling laws analysis.
>
> |       | MMStar | MME   | SEEDBENCH_II | ScienceQA | MathVista | AI2D   |
> |-------|--------|-------|--------------|-----------|-----------|--------|
> | SOLO  | 88.75  | 92.36 | 98.59        | 97.75     | 84.7      | 96.55  |
> | mLLAVA| 40.32  | 92.52 | 92.89        | 89.25     | 62.73     | 83.29  |
>
>
> [1] Training Compute-Optimal Large Language Models. Jordan Hoffmann et al
>
> [2] Scaling Laws for Neural Language Models. Jared Kaplan et al

---

> > ### Author Response · Authors · 2024-09-13
> >
> > **Challenges in large-scale training and deployment**:  We conduct a detailed analysis to measure the inference speed of SOLO compared to several LVLMs with heterogeneous architectures, including LLaVA-Next, LLaVA-Interleave, and Qwen-VL. The evaluation is conducted on a single A100 GPU using 10,000 COCO images and a consistent prompt:  “Generate the Caption for the <image>”. All inference code is implemented using HuggingFace, and the inference latency is measured from from the input being provided to the model and the output of the first token for fair comparison among all models since they may generate free-form responses in different lengths. The results, shown below, indicate that SOLO consistently outperforms other LVLMs with heterogeneous architectures by a significant margin, demonstrating its clear advantage in inference speed.
> > | Model              | Inference Latency (second) |
> > |--------------------|-------------------|
> > | LLaVA-Next         | 0.0641            |
> > | LLaVA-Interleave   | 0.0691            |
> > | Qwen2-VL           | 0.0588            |
> > | SOLO               | 0.0235           |
> >
> > In addition, we also measure the training speed of SOLO compared to LVLMs with heterogeneous architectures by measuring throughput. Using the same 8xA100 server, we compare the number of tokens processed per second during SOLO training and the official LLaVA implementation. Our SOLO implementation achieves a throughput of 20K tokens per second, while LLaVA reaches 10.5K tokens per second, highlighting SOLO's significant advantage in training speed.
> >
> > For deployment, the heterogeneous architecture can hamper large-scale services. Existing specialized AI chips, like [1], are mostly optimized for the single Transformer architecture, and state-of-the-art inference libraries, such as vLLM [2] and MLC-LLM [3] are also specially designed for the single Transformer. These present significant challenges in the deployment of these heterogeneous-architectured LVLMs on end devices.
> >
> > [1] Techcrunch. Etched is building an AI chip that only runs one type of model. https://techcrunch.com/2024/06/25/etched-is-building-an-ai-chip-that-only-runs-transformer-models/.
> >
> > [2] Efficient memory management for large language model serving with pagedattention. Woosuk Kwon et al
> >
> > [3] MLC team. MLC-LLM, 2023. URL https://github.com/mlc-ai/mlc-llm.

---

> ### Author Response · Authors · 2024-09-13
>
> > Insufficient explanation in FIgure 8
>
> Thank you for the feedback regarding Figure 8. We clarify the experimental setting and results explanations as follows.
>
> **Experimental setting**: We conducted a controlled analysis of the instruction fine-tuning stage, comparing SOLO with mLLaVA, both trained using SOLO's recipe. For evaluation, we fine-tuned both models for 50 steps, as their pre-trained versions were not effective at following instructions. Given that both models are fine-tuned on the same dataset, we can directly compare their average improvement in benchmark performance per training token during training. In implementation, we measured performance improvement every 500 steps, normalized by the number of training tokens encountered during those steps, and averaged this to calculate the final metric.
>
> **Experimental results**: Figure 8 shows that SOLO outperforms mLLaVA on the "performance improvement per token" metric across all evaluation benchmarks. This suggests SOLO benefits more from high-quality instruction fine-tuning data and demonstrates better scalability, indicating that its performance could further be improved more compared to mLLaVA with more data.
>
>
>
>
>
> > Limitations should be described
>
> Thanks for the suggestion. We address the limitations of SOLO in the final section, particularly its performance compared to state-of-the-art (SOTA) LVLMs. Regarding concerns about SOLO's computational costs, we wish to emphasize that it was trained with limited academic resources—specifically 8 A100 GPUs over a period of around 10 days. This contrasts sharply with the significantly larger resources used for training SOTA LLaVA-style LVLMs of similar size (~7B). For instance, in the DeepSeek-VL technical report [1], the authors mentioned that DeepSeek-VL-7B was trained on a cluster of 64 nodes for 5 days, each comprising 8 Nvidia A100 GPUs, totaling 512 GPUs—this represents a resource scale 64 times greater than that used for SOLO.
>
> [1] DeepSeek-VL: Towards Real-World Vision-Language Understanding. Haoyu Lu et al

---

> ### Author Response · Authors · 2024-09-18
>
> Hi Reviewer,
>
> Please accept our sincere gratitude for all your suggestions on our work. we hope our responses have answered your questions. We are happy to discuss any questions you may have at this stage!
>
> Best, Authors

---

### Decision · Action_Editor_e8Z5 · 2024-10-25

**Recommendation:** Accept with minor revision

**Comment:**

I think the overall contribution of this paper is sufficient to be published as a TMLR paper. However, there are some minor issues that need a minor revision.

1. Clarity: One main issue of the current version is clarity. I don't mean clarity in terms of grammar or fluency, but I mean clarity in terms of terminology. For example, as many reviewers pointed out, this paper uses "scalability" in various meanings. I don't think "an architecture can demonstrate sustained performance improvement through scaling (e.g., more computational resources and/or data) without hitting any inherent bottleneck in the machine learning system" is a rigorous definition. What is "inherent bottleneck in the system"? How is the bottleneck defined? Why do the existing methods have the bottleneck? If they have the bottleneck, can we actually observe the performance degeneration by scaling up? I encourage the authors to polish the manuscript to enhance readability and clarity.
2. The claims may need additional refinement. For example:
    - Constrained visual capabilities: It is not well-supported. If this claim is true, then one might observe any flaw of the existing heterogenous VLMs in terms of "performance". However, this paper does not show the performance limitation of heterogenous VLMs. It it is impossible to support this empirically, I'd like to recommend removing this from the main claim. It could be described in "discussion".
    - Challenging large-scale training and deployment: Inference speed is not the only bottleneck in training and deployment. This argument might be clarified by "better inference speed". If inference speed is not the only reason, then it should be empirically supported as the other claims.
3. I don't think the "Appendix" of this paper is not optional. Since TMLR does not have a page limitation (it is only used for deciding the review period), I strongly recommend moving Appendix contents to the main paper, especially for Section G.

**Audience:**

I believe seeking a better architecture for vision-language models is an important issue. There might be many audience interested in the findings of this paper.

**Claims And Evidence:**

### Claims

- This paper introduces Scalable Vision-Language Modeling (SOLO) to tackle four bottlenecks of existing VLM relying on a pre-trained visual encoder
    - Constrained visual capabilities (compared to LLM)
    - Challenging large-scale training and deployment (due to the difference between visual encoder and textual encoder architectures)
    - Complexity in scaling analysis due to multiple components (because there are three different components, including the visual encoder, the connector, and the LLM. Therefore, unlike LLMs, it is difficult to analyze scaling laws for LVLMs)
    - Limited image pre-processing flexibility (because of the dependency on the CLIP-ViT-L/14 @336px model => We need to use a squared image with the fixed resolution)
- SOLO tackles these issues by employing a single Transformer for unified end-to-end vision-language modeling.

### Evidence

- The proposed architecture is solely based on LLM (Mistral) rather than using multiple encoders (e.g., separate Vision Transformer and Language Transformer).
- In the revised paper, the authors supported their claims with additional experiments.
    - Constrained visual capabilities: This claim is not well supported. There are some related discussions (e.g., Section 2), but I couldn't find more rigorous evidence, such as experimental results or conceptual explanations.
    Challenging large-scale training and deployment: The revised paper's Table 3 shows that SOLO has a faster inference speed. However, I think inference speed is not the only challenge during training and deployment. It would be better to tone down this claim.
    Scaling analysis: The revised paper's Table 4 shows that SOLO is more predictable than mLLAVA in terms of scaling analysis.
    - Image pre-processing flexibility: The revised paper's Table 5 shows that when using larger image resolution, SOLO can perform better than the other methods.

---

> ### Author Response · Authors · 2024-11-13
>
> Dear AE,
>
> Thanks for your appreciation of our work! We have updated the camera-ready version with the latest revision. We summarize our responses and revisions as follows, according to your comments:
> 1. **Clarity issues**
>
> Thanks for pointing this out! We polish the definition of “scalability” in the introduction as: We define an architecture as scalable when it exhibits consistent performance improvements as computational resources and training data increase, maintaining a positive scaling law relationship up to practical limits. This is in contrast to architectures that show diminishing returns or performance plateaus at larger scales due to fundamental architectural limitations such as information bottlenecks (i.e., structural limitations in information transmission) in visual perception.
>
> In this definition, we formally define the inherent bottleneck as the structural constraints that limit information transmission. We explain that existing LVLMs encounter this bottleneck due to limitations in their visual perception capabilities. We provide empirical evidence explained in the subsequent response that illustrates the resulting diminishing returns.
>
> 2. **The claims may need additional refinement**
>
> Thanks for pointing it out! We refine our paper based on these two points:
> - Our analysis in the first part of Section 6.2, 'Superior Scaling Properties of SOLO' provides evidence demonstrating the performance limitations of heterogeneous LVLMs, compared to SOLO.  Specifically, we show that LLaVA-Style LVLMs show diminishing returns when trained on high-quality instruction fine-tuning data compared to SOLO. In our revision, to enhance clarity, we now begin this analysis with a concise statement: “We demonstrate that existing LVLMs with heterogeneous architectures exhibit diminishing returns despite increases in high-quality instruction fine-tuning data while SOLO shows better scaling behavior.”
> - We refine this argument in the introduction and Section 2 by focusing our claim on computational efficiency in terms of training and inference speed.  In addition, we also add the training speed comparison in the second part of Section 6.2 to support our argument.
>
> 3. **Appendix content**
>
> Thank you for your feedback! We add a new discussion section to address several compelling findings that were previously in the Appendix. This section explores three key observations: (a) the disconnect between vision-language pre-training loss or instruction fine-tuning loss and actual model performance, (b) the challenges in achieving an optimal balance between visual and textual capabilities during pre-training, and (c) a comparative analysis of scaling laws between the two model types. We present supporting experimental results and detailed analysis for each of these points.
>
>
> **If you have further comments or requested revisions, please let us know! Thanks for your valuable feedback!**
>
> Authors of Paper 3133